# The paleoclimatic footprint in the soil carbon stock of the Tibetan permafrost region

Jinzhi Ding [1], Tao Wang[1,2,3], Shilong Piao [1,2,4,5], Pete Smith [6], Ganlin Zhang[7], Zhengjie Yan[3], Shuai Ren[8], Dan Liu[1], Shiping Wang[1], Shengyun Chen[9], Fuqiang Dai[10], Jinsheng He [11], Yingnian Li[12], Yongwen Liu [1,2], Jiafu Mao [13], Altaf Arain[14], Hanqin Tian [15], Xiaoying Shi[13], Yuanhe Yang [16], Ning Zeng [17] & Lin Zhao[18]

Tibetan permafrost largely formed during the late Pleistocene glacial period and shrank in the Holocene Thermal Maximum period. Quantifying the impacts of paleoclimatic extremes on soil carbon stock can shed light on the vulnerability of permafrost carbon in the future. Here, we synthesize data from 1114 sites across the Tibetan permafrost region to report that paleoclimate is more important than modern climate in shaping current permafrost carbon distribution, and its importance increases with soil depth, mainly through forming the soil's physiochemical properties. We derive a new estimate of modern soil carbon stock to 3 m depth by including the paleoclimate effects, and find that the stock ($36.6^{+2.3}_{-2.4}$ PgC) is triple that predicted by ecosystem models (11.5 ± 4.2 s.e.m PgC), which use pre-industrial climate to initialize the soil carbon pool. The discrepancy highlights the urgent need to incorporate paleoclimate information into model initialization for simulating permafrost soil carbon stocks.

[1] Key Laboratory of Alpine Ecology, Institute of Tibetan Plateau Research, Chinese Academy of Sciences, Beijing 100101, China. [2] CAS Center for Excellence in Tibetan Plateau Earth Sciences, Chinese Academy of Sciences, Beijing 100101, China. [3] School of Life Sciences, Lanzhou University, Lanzhou 730000, China. [4] Sino-French Institute for Earth System Science, College of Urban and environmental Sciences, Peking University, Beijing 100871, China. [5] University of Chinese Academy of Sciences, Beijing 100049, China. [6] Institute of Biological and Environmental Sciences, School of Biological Sciences, University of Aberdeen, Aberdeen AB24 3UU, UK. [7] State Key Laboratory of Soil and Sustainable Agriculture, Institute of Soil Science, Chinese Academy of Sciences, Nanjing 210008, China. [8] Shenzhen Key Laboratory of Circular Economy, Shenzhen Graduate School, Peking University, Shenzhen 518055, China. [9] State Key Laboratory of Cryosphere Science Northwest Institute of Eco-Environment and Resources Chinese Academy of Sciences, Lanzhou, Gansu 730000, P.R. China. [10] School of Tourism and Land Resources, Chongqing Technology and Business University, Chongqing 400067, China. [11] College of Urban and Environmental Sciences, Peking University, Beijing 100871, China. [12] Key Laboratory of Adaptation and Evolution of Plateau Biota, Northwest Institute of Plateau Biology, Chinese Academy of Sciences, Xining, Qinghai 810008, China. [13] Environmental Sciences Division, Climate Change Science Institute, Oak Ridge National Laboratory, Oak Ridge, TN 37831, USA. [14] School of Geography and Earth Sciences and McMaster Centre for Climate Change, McMaster University, Hamilton, Ontario, Canada. [15] International Center for Climate and Global Change Research, School of Forestry and Wildlife Sciences, Auburn University, Auburn, AL 36849, USA. [16] State Key Laboratory of Vegetation and Environmental Change, Institute of Botany, Chinese Academy of Sciences, Beijing 100093, China. [17] Department of Atmospheric and Oceanic Science and Earth System Science Interdisciplinary Center, University of Maryland, College Park, MD 20742, USA. [18] Cryosphere Research Station on Qinghai–Xizang Plateau, State Key Laboratory of Cryospheric Science, Northwest Institute of Eco–Environment and Resources (NIEER), Chinese Academy of Sciences, Lanzhou 730000, China. Correspondence and requests for materials should be addressed to T.W. (email: twang@itpcas.ac.cn)

There is evidence that the paleoclimate has influenced the cycling of soil carbon through shifting biomes[1–3] and by altering soil physiochemical properties[4]. If such influences were common, then the current distribution of soil carbon stocks should contain footprints of the paleoclimate at timescales ranging from centuries to millennia[3]. Such paleoclimate signals would be expected to be strongest in permafrost soils, where much of the soil carbon is locked in a frozen state[5–7], and, therefore, only susceptible to change from the most extreme paleoclimate events. Only the warmest and coldest periods are likely to leave recognizable changes on the soil carbon in these soils.

Land regions of permafrost constitute the largest soil carbon pool in terrestrial ecosystems[8,9] and, with an area of 1.06 million km$^2$, the Tibetan Plateau is the largest alpine permafrost area outside the polar regions[10]. Permafrost carbon cycling in the Tibetan Plateau has, therefore, come under intense scrutiny[11]. Currently, the region is characterized by a semiarid climate[12,13], and has a rate of warming of twice the global average[14]. Evidence from preserved relict permafrost and periglacial phenomena, indicate that the main body of the existing Tibetan permafrost was formed during the last glaciation period, at the end of the late Pleistocene[15,16], which was characterised by cold and arid periglacial environments. Subsequently, the Tibetan permafrost experienced intensive and extensive degradation in the middle Holocene, during which time the temperature increase was greater than that during any of the following warming periods, and the total permafrost area was reduced to ~50–60% of the current area[16]. To date, it has not been demonstrated whether these extreme paleoclimate signals are retained in the Tibetan permafrost soil carbon, or if they have already been erased by subsequent smaller climate oscillations. Understanding the influence of past climate extremes on Tibetan soil carbon could shed light on the vulnerability of the permafrost soil carbon pool to future climate change. This is especially so for

the case of the mid-Holocene, which could provide a geological analogue for future climate over the Tibetan Plateau[17].

Here, we compile data from 1114 sites collected during 11 field campaigns across the Tibetan permafrost region, and use multiple statistical techniques to assess the relative importance of climates from the last glacial maximum (LGM), an extremely cold period 22,000 years ago; the mid-Holocene (MidH), a hypsithermal period about 6000 years ago; and the modern climate (1975–2015) in driving the current spatial pattern of permafrost soil carbon. Our results show that paleoclimate is more important than the modern climate in determining current soil carbon stocks, and its importance increases progressively with soil depth. Direct physical explanations for this pattern are offered. When we include paleoclimate as an additional predictor in a machine learning algorithm to re-assess Tibetan soil carbon stocks in the top three metres of the soil, we find that the present generation of terrestrial ecosystem models are biased towards low soil carbon stocks, highlighting the need to include paleoclimate information in soil carbon model initialization.

## Results

**Paleoclimate controls on permafrost soil carbon distribution.** Based on the compiled soil carbon data set for the Tibetan permafrost region (Fig. 1), we used random forest modelling to rank the relative importance of paleoclimate (LGM and MidH) and modern climate in driving the spatial pattern of permafrost carbon in the top 30 cm of the soil. According to the increase in the residual sum of squares (see Methods), the random forest models indicated that paleo-temperature in LGM and MidH was almost as important as modern precipitation (Fig. 2a). Furthermore, variation partitioning modelling showed a larger contribution from paleoclimate alone (12%) than from modern climate alone (4%) in predicting soil carbon distribution. Paleoclimate and

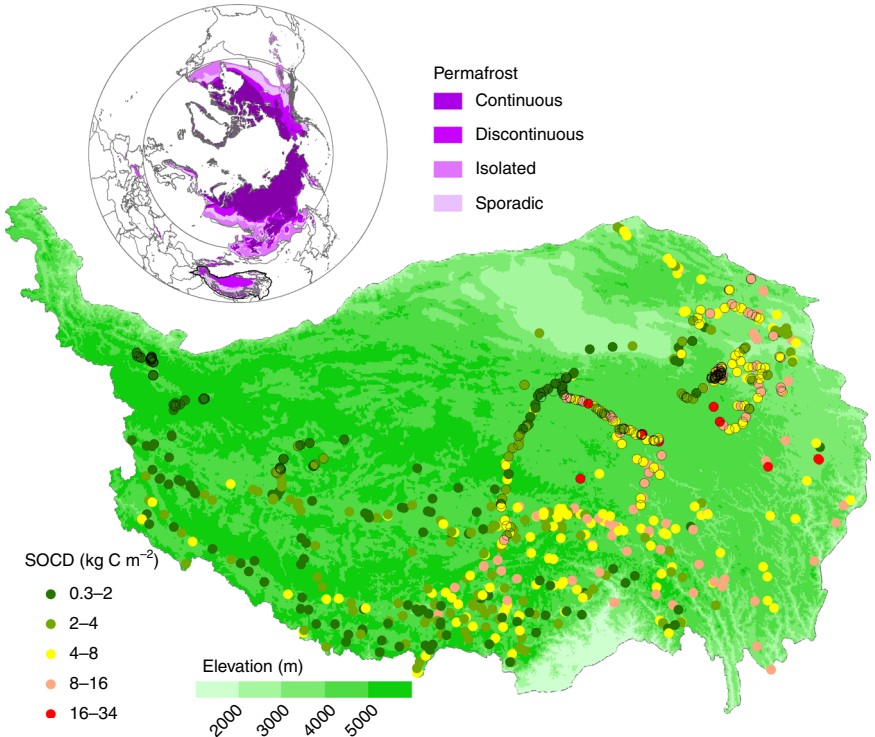

**Fig. 1** The locations and soil organic carbon density (SOCD) of the top 30 cm layer for the 1114 sampling sites over the permafrost regions on the Tibetan Plateau. The deep soil carbon measurements (at a depth of more than 2 m) are indicated by black outlines for the coloured dots. The modern permafrost map was obtained from the National Snow & Ice Data Center[65]

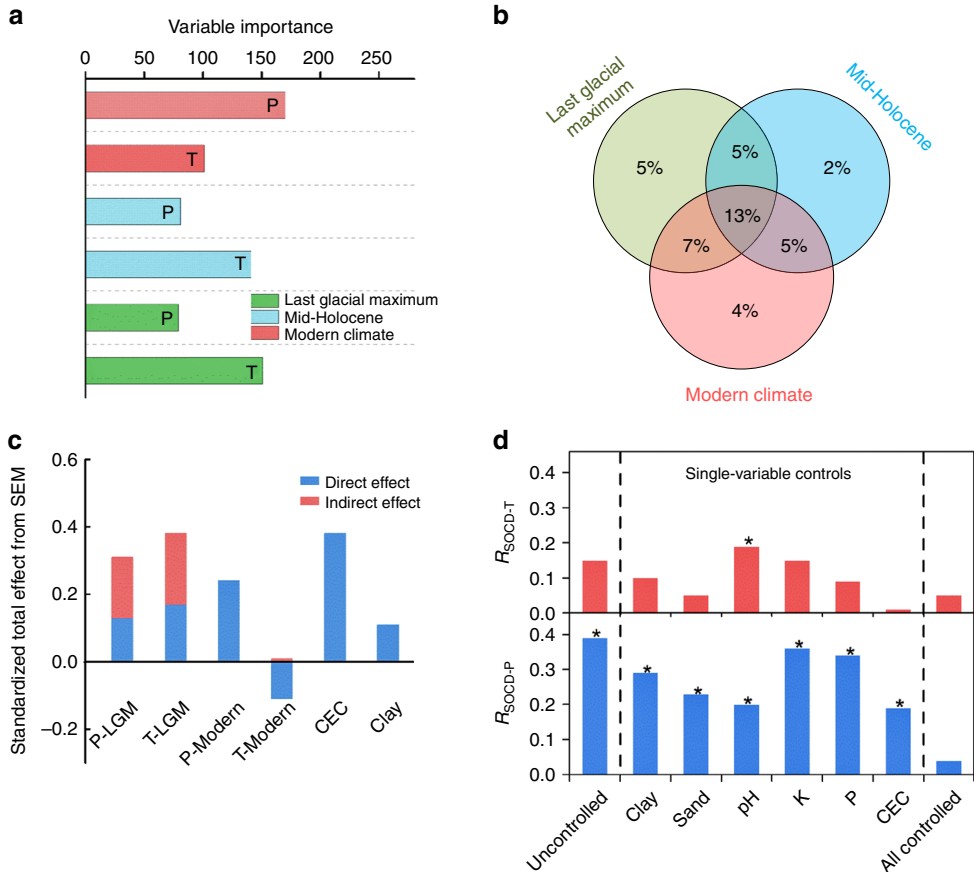

**Fig. 2** Relative importance of paleo- and modern climates in regulating soil carbon stock over the permafrost regions of the Tibetan Plateau based on multiple statistical models. The relative importance matrix includes results from: **a** Random Forest modelling; **b** Variation Partitioning modelling; **c** Structural Equation Modelling (SEM). T, the first principle component of all temperature-related variables; P, the first principle component of all precipitation-related variables. **d** shows the partial correlation coefficients between SOCD of the top 30 cm layer and mean annual temperature ($R_{\text{SOCD-T}}$) and mean annual precipitation ($R_{\text{SOCD-P}}$) in modern times when none of, each of, and all of the soil property variables were controlled (separated by the vertical broken lines). The soil property variables are clay content (Clay), sand content (Sand), soil pH, total potassium (K), total phosphorus (P) and cation exchange capacity (CEC). *Denotes significant at $P < 0.05$

modern climate jointly explained an additional 25% of the total variance (Fig. 2b). These results suggest that paleoclimate has a strong influence on soil carbon distribution, even in the top 30 cm soil layer. Paleo-temperature was found to be the most important paleoclimate variable for the formation of the current permafrost soil carbon distribution (Fig. 2a). This finding is as expected, because the areal extent of permafrost was mainly shaped by low temperatures during cold periods such as the LGM (Supplementary Fig. 1). The compartmentation of soil carbon in the remnants of ancient buried permafrost formed during paleoclimates may greatly influence current soil carbon stocks. This finding, however, differs from the results for arid and semiarid regions, where paleo-precipitation was found to be the main driver among all the investigated factors, including both paleo and modern climates[3].

There are two mechanisms by which paleoclimate could influence current soil carbon (Supplementary Fig. 2). In the first mechanism, the organic carbon in modern soils is directly derived from the vegetation formed under paleoclimate conditions[3]. In the second mechanism, the paleoclimate exercises a degree of control over the formation of soil physiochemical properties, which subsequently, indirectly, determine the degree of stabilization of the organic carbon[4]. We have used structural equation modelling to evaluate these direct and indirect effects of paleoclimate on the current distribution of soil carbon in the

top 30 cm of the soil (Fig. 2c; Supplementary Fig. 3, Supplementary Table 1). Direct effects on permafrost soil carbon distribution were observed for both temperature (standardized path coefficient = 0.17, $P < 0.001$) and precipitation (standardized path coefficient = 0.13, $P < 0.001$) in the LGM (Supplementary Fig. 3). This observation is in agreement with the results of soil radiocarbon ($^{14}$C) dating studies in the relict permafrost, where the carbon age can be dated to 20~40 thousand years BP[16,18].

The structural equation modelling results also reveal a distinct and indirect impact of paleoclimate on soil carbon through changing soil physiochemical properties (Fig. 2c, Supplementary Fig. 3). The soil physiochemical characteristics (i.e., soil texture, cation exchange capacity, total phosphorus and potassium), determining the capacity to stabilize soil carbon inputs[19–21], have evolved slowly under the influence of past climate regimes, and there is increasing evidence showing the importance of physiochemical properties in controlling soil carbon stock over the Tibetan Plateau[22,23]. This is supported by other studies, which have found that considering a diversity of soil evolution processes was a key factor in assessing soil carbon or nitrogen patterns[24,25].

Our results further show that the indirect effect (standardized effect = 0.39) is larger than the direct effect (standardized effect = 0.30; Fig. 2c), suggesting that the influence of paleoclimate on Tibetan permafrost soil carbon distribution operated primarily through modifying soil physiochemical properties, with

the direct effect taking a secondary role. If the preservation of paleo-vegetation signals in the upper soil layers predominantly explained the current soil carbon distribution, the legacy impacts of the LGM should be significantly concealed by those of the MidH. During the Holocene Thermal Maximum (HTM), intensive permafrost thawing occurred down to a depth of ~15–25 m on the Tibetan Plateau[16,26], and the areal extent of permafrost represented by the freezing index (see Methods) also decreased substantially, with the magnitude of the mean decline being nearly 22% over the Tibetan Plateau (Supplementary Fig. 4). Such thawing would be expected to have greatly increased soil carbon decomposition. However, the fact that we can detect the LGM legacies on the upper-layer soil carbon, means that the influence of paleoclimate on soil properties is the most likely explanation of paleoclimate legacies on current soil carbon stocks[4]. The indirect mechanism is further supported by radio-carbon dating of soil organic carbon in the soils over the Tibetan Plateau (Supplementary Table 2). The relatively young age obtained by this method suggests that the current top soil layers are not likely to originate from LGM-vegetation. The results of correlation analysis between soil carbon and modern climate variables, with and without controlling for the effect of soil physiochemical properties, gives further robustness to the above findings derived from the structural equation modelling. Specifically, the correlations significantly decreased, or even became insignificant, after removing the effects of soil properties (Fig. 2d), confirming that the paleoclimate effect mainly operated through changing soil properties.

To quantify the relative contributions of paleoclimate and modern climate to current soil carbon distributions at different soil depths, the Lindeman–Merenda–Gold method was used (see Methods). The results suggested an increasingly important role of paleoclimate with increasing soil depth (Fig. 3). Specifically, in the top 10 cm of the soil, the variances explained by paleoclimate (LGM and MidH) and modern climate were 58% (25 and 33%) and 42%, respectively, and at 50 cm were 72% (38 and 34%) and 28%, respectively. At soil depths greater than 200 cm, the current

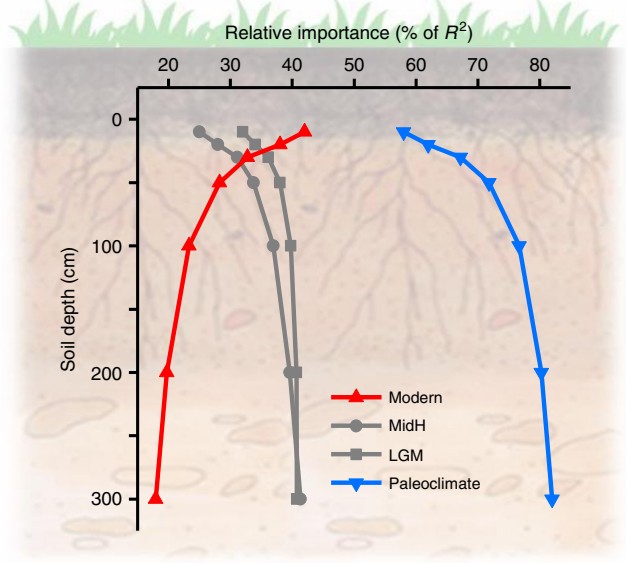

**Fig. 3** Standardized relative importance of paleo- and modern climate for the spatial pattern of soil carbon stock over the permafrost regions on the Tibetan Plateau at various soil depths. The relative importance was determined by the Lindeman–Merenda–Gold method[60]. Note that the relative importance of paleoclimate (blue curve) is the sum of the values of LGM and MidH (grey curves) by layer

soil carbon distribution is predominantly regulated by paleoclimate (more than 80% of the relative variance) (Fig. 3). This vertical pattern was not found when a global data set, mainly from non-permafrost regions, was analysed, although paleoclimate did explain a large proportion of the variation of carbon distribution in the top soil layer[3]. This result is expected, since organic carbon in surface soil layers is much more vulnerable to microbial decomposition in response to warming-induced top-down thawing, than in deeper layers of permafrost[9]. In addition, the stronger effect of physical protection and chemical sorption in deeper soils should also play a role[20,27].

**A new estimate of Tibetan permafrost soil carbon stocks.** Previous assessments of the Tibetan soil carbon pools have relied on a collection of predictors based only on modern climate and remote sensing-based vegetation features[28–31]. Here, we have merged modern climate and remote sensing-based methods common in previous estimates, with paleoclimate, landform and soil geochemical properties (Supplementary Fig. 5) in multiple machine learning algorithms, to make a new estimate of the permafrost soil carbon pool over the Tibetan Plateau (see Methods).

According to the results from the best predictive model, Support Vector Machine (Supplementary Fig. 6), the Tibetan soil organic carbon pool to a depth of 3 m (see Methods) was estimated to be 36.6 PgC (95% confidence range: 34.2–38.9 PgC), and the mean soil organic carbon density (SOCD) was estimated to be 15.4 kg C m$^{-2}$ (95% confidence range: 14.4–16.4 kg C m$^{-2}$) (Supplementary Fig. 7). It is noteworthy that the actual soil layer thickness could be highly variable over the Tibetan Plateau, but has been assumed to be uniform in previous estimates[28–31]. We included the soil layer thickness in the machine learning-based model, and found that lack of information on the spatial variation of soil layer thickness could have led to an overestimation of the 3 m soil carbon stock by 3.6 PgC ( ~10%).

**Model-observation comparison of Tibetan soil carbon stock.** The comparison of our new estimate of permafrost soil carbon stock with estimates from state-of-the-art terrestrial ecosystem models suggests that, generally, the models have underestimated soil carbon stock over the Tibetan Plateau (Fig. 4a). The LPJ-wsl and TEM6 models are exceptions to this general rule. The significant variability in the performance of the ecosystem models could be related to differences in the models' representation of soil carbon input and output (see Supplementary Discussion). Here, we used the Bayesian model averaging method (BMA), which is conditional on an independent observation data[32], to tone down the role of models that have notable deficiencies in representing major physiological processes. Given the availability of satellite-derived net primary productivity (NPP) product[33], we adopted NPP, as an indicator of vegetation carbon input, to rank the model performance. Larger weights were assigned to models that have a better performance in simulating NPP with respect to satellite-derived observations. We found that the pool size of the weighted ensemble mean of the 11 models is 11.5 ± 4.2 s.e.m PgC, which is less than one-third of our new estimate. In addition, the models fail to describe the spatial pattern of soil carbon stock. The best spatial correlation between the modelled stock and our estimate was found for ISAM ($r = 0.55$, $P < 0.001$), and GTEC ($r = 0.45$, $P < 0.001$), while the correlations for CLASS_CTEM, LPJ_wsl, TEM6 and VEGAS2.1 are less than 0.3 (Fig. 4b). We further calculated the spatially explicit indices relative distance (RD) and cross correlation (CC) at multiple scales to examine spatial similarities between the models and our estimate using the Comparison Map Profile (CMP) method[34]. We found

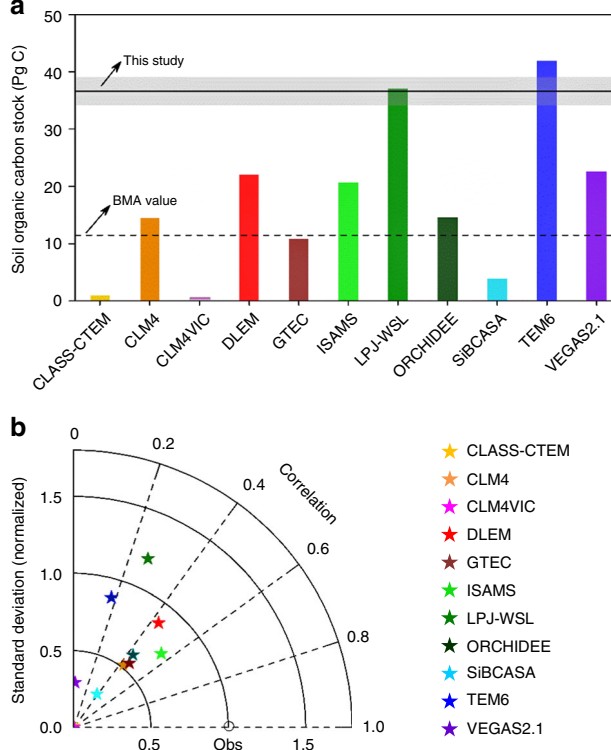

**Fig. 4** Comparison of soil carbon stock simulated by 11 ecosystem models with estimates from this study. **a** Shows total soil organic carbon stock. BMA value represents the weighted ensemble mean of the model outputs based on the Bayesian model averaging method[31]. **b** is the Taylor diagram which shows correlation coefficients between the gridded model simulations and estimates from this study (Obs), and the normalized standard deviation of model simulations. The model simulations originated from the Multi-scale Synthesis and Terrestrial Model Intercomparison Project (MsTMIP)[64]

that eight out of eleven models, particularly CLASS_TEM, CLAM4VIC, GTEC and SiBCASA, clearly underestimated the Tibetan soil carbon stock (Supplementary Fig. 8). We also observed a large spatial inconsistency between simulated soil carbon stock and our estimates, with a low correlation in most parts of the study area (Supplementary Fig. 9). Even the models such as LPJ-WSL and TEM6, which simulated a total soil carbon stock of comparable size to our estimate, still fail to capture the spatial distribution of soil carbon stock, as indicated by the generally low values of correlation (Supplementary Fig. 9). For these two models, the overestimation in the western Tibetan Plateau effectively compensates for the underestimation in the east, leading to an apparent model-observation match in terms of the total soil carbon stock (Supplementary Fig. 8).

There was a nearly perfect relationship between the initial soil carbon stock after model spin-up and the present-day (1980–2010) stock across the models ($r^2 = 0.99$, $P < 0.001$; Supplementary Fig. 10), highlighting the critical role of model spin-up in the estimation of soil carbon stock. The present-day soil carbon stock was derived from the transient simulation that started from steady-state initial conditions after spin-up and was run forward in time through the historical period until 2010, using observed time-varying climate and $CO_2$[35]. In these models, the soil carbon pools were initialized by the early 1900's climate rather than the paleoclimate. The fact that paleoclimate information is not used for soil carbon initialization may have led to significant errors in the model estimates of soil carbon

stocks. To test whether the lack of paleoclimate information led to the biased simulation of modern carbon stock by the ecosystem models, we correlated the paleoclimates to the difference between the ensemble model simulations (paleoclimate not considered) and our estimation (paleoclimate considered). The results reveal significant and strong correlations with paleo-temperature ($r = -0.21$, $P < 0.001$ for LGM; $r = -0.22$, $P < 0.001$ for MidH) and weak correlations with paleo-precipitation ($r = -0.03$, $P < 0.001$ for LGM; $r = -0.09$, $P < 0.001$ for MidH). These results are consistent with the relative importance analysis that shows greater importance of paleo-temperature in regulating the current soil carbon stock (Fig. 2a). This result is expected because the formation of the permafrost due to low temperature may lead to the inhibition of both respiration in frozen soils and vertical mixing of soil carbon between the surface and permafrost layers. These processes are generally lacking in the current generation of ecosystem models[36]. This study provides evidence that illustrates, for the first time, the bias caused by the lack of paleoclimate information in ecosystem models.

In addition to paleoclimates, other processes that are not well resolved in current ecosystem models, such as soil carbon turnover time, may also account for some of the model underestimation. Firstly, poor model representation of the long turnover time of deep soil carbon, especially in permafrost-affected regions, may lead to significant underestimation of soil carbon stock[37,38]. The current soil carbon models adopt a single vertically integrated soil carbon pool, without considering the vertical gradients in soil carbon stability and decomposability[37]. In reality, the part of the organic carbon stored in deep layers, many thousands of years older than the surface organic carbon, is generally considered to be stable due to low decomposition rates, especially in permafrost-affected regions[7,9]. Therefore, it's possible that the omission of this vertical dimension in the modelling of soil carbon cycling may contribute to the models' underestimation. This hypothesis is in agreement with our model-observation comparison analysis, which showed that there was a higher degree of underestimation in permafrost-affected soils than in non-permafrost affected soils (Supplementary Fig. 11). Secondly, several typical alpine vegetation types, such as marsh meadow and alpine meadow, which are characterized by relatively high organic carbon density and slow soil carbon turnover rates, were not well represented in the models of the MsTMIP protocol[39]. To test whether this limitation contributes to the model underestimation, we compared the relative distance between the observed and modelled soil carbon stock for different vegetation types. We found that the soil carbon stock was severely underestimated in marsh meadow (mean relative distance = −88%) and alpine meadow (mean relative distance = −64%) (Supplementary Fig. 11). Since these two types of alpine vegetation cover about one-third of the total area of the plateau, accounting for 41% of the total soil carbon stock, these underestimates represent a major contribution to the overall model underestimation.

## Discussion
We compiled a unique soil carbon data set to show that extreme paleoclimatic signals from the LGM and MidH can be detected in the current soil carbon stock across the Tibetan permafrost regions. The paleoclimatic legacies are retained mainly through impacts on soil physiochemical properties, particularly in the upper soil horizons. We also found a clear increasing trend of paleoclimate effects on current soil carbon stock with soil depth. These findings emphasize the necessity of considering paleoclimate legacies, especially paleo-temperature conditions, when estimating contemporary alpine permafrost carbon stocks. This is

of critical importance for improving assessments of permafrost carbon stocks, particularly in deep layers, where a greater influence of paleoclimate was observed. Our new estimate of the carbon pool, obtained by including paleoclimate as an additional predictor, is triple the size of current modelled values. Future modelling of soil carbon cycling should include paleoclimate as well as its interaction with soil properties during the model spin-up period so as to accurately represent the impacts of paleoclimate on soil properties. In addition, the methodology introduced in this paper could be used to quantitatively assess the paleoclimatic footprint on the permafrost soil carbon stock in other permafrost regions such as the pan-arctic region. Such assessments could provide a more complete understanding of paleoclimate effects on permafrost soil carbon stock that in turn can help with the understanding of permafrost soil carbon dynamics in a warmer future.

However, we should caveat our findings regarding the impact of paleoclimate on soil carbon distribution, because the paleoclimate parameters are model-derived and not validated from proxy data. There is a growing discipline of reconstructing past climate records based on a variety of paleoclimatic proxies such as lake sediments, pollen and glaciers over some regions of the Tibetan Plateau[40–42]. Further studies which blend paleoclimatic model simulations with such multiproxy data, along with their reasonable climatic interpretation, are required so as to provide spatial reconstructions of observationally-constrained paleoclimate over the Tibetan Plateau. Additionally, to enable future land surface models to fully account for paleoclimatic impacts on permafrost soil carbon stock, other paleoclimatic variables such as humidity, pressure and radiation, not just temperature and precipitation, should be prepared and downscaled to the relatively high temporal resolution (e.g., sub-daily) required by the models.

## Methods

**Study area and soil carbon data collection.** The Tibetan Plateau, the highest and largest plateau in the mid-latitudes, has the world's largest area ($1.06 \times 10^6$ km$^2$) of alpine permafrost, accounting for 8% of the Northern Hemisphere's permafrost, and 75% of its alpine permafrost[10,23]. Due to strong orographic effects, there is a great spatial variability of many environmental conditions, such as altitude, volumetric soil water content and soil layer thickness, over the Tibetan Plateau[13,43].

In addition to low temperature, the Tibetan climate is characterized by aridity in the main body of the plateau[13], and so, due to limited water content in the soil, the periglacial landforms and processes typical of the polar regions, such as ground heave, subsidence and ice wedges, are less developed here[12]. According to in situ observations, the ice content in Tibetan permafrost is 12%[44] and the active layer thickness ranges from 0.6 to 3.5 m[45]. The Tibetan Plateau is mainly covered by cold- and drought-adapted vegetation, including alpine steppe and alpine meadow communities, accounting for 63% of the total area of the plateau, with the remaining parts covered by forests, shrubs, alpine deserts, marsh meadows and cultivated lands. Cambisols and leptosols are the two main soil types on the plateau, together covering about 77% of the whole area[30].

We have synthesized soil data collected from 1114 sites during 11 field campaigns conducted by multiple research teams over the past three decades. The data sources include China's second national soil survey (National Earth System Science Data Sharing Infrastructure, National Science & Technology Infrastructure of China (http://www.geodata.cn), published work[23,29–31,46–49] and some unpublished data. This data set represents the state-of-the-art soil carbon data set for the Tibetan Plateau (Fig. 1). The spatial representativeness of the sampling sites, in terms of both sample size and spatial coverage, is unprecedented in the region.

Soil organic carbon density (SOCD, kg C m$^{-2}$) was estimated based on soil organic carbon concentration (SOCC, g kg$^{-1}$), bulk density (BD, g cm$^{-3}$), soil layer depth (T, cm) and rock (diameter larger than 2 mm) content (C, %) in layer $i$, using the following formulation:[30]

$$SOCD = \sum_{i=1}^{n} T_i \times BD_i \times SOCC_i \times \frac{(1 - C_i)}{100} \quad (1)$$

Since all sites have data for the top 30 cm soil layer, we first assessed the relative contributions of paleo- and modern climate using 30 cm SOCD. The missing values of BD (21%) were estimated using pedotransfer functions between SOCC and BD, while missing rock content values (29%) were estimated by using the mean values for the same soil type.

The full data set includes 325 sites with deep soil data (extending to more than 200 cm in depth) (Fig. 1), with SOCD data for seven different layers: 0–10, 10–20, 20–30, 30–50, 50–100, 100–200 and 200–300 cm. These data substantially improved the spatial coverage, especially in the western half of the permafrost regions, where the number of deep soil data per unit area was less than half of that in the eastern half in a previous study[30]. These deep soil carbon data enabled us to develop a function of SOCD changes with soil depth, which was subsequently used to extrapolate deep soil carbon stock values for the sites without deep soil carbon records.

**Climate data.** We compiled a climate data set composed of climate over the period 1975–2015 (modern climate), and paleoclimates in the mid-Holocene (MidH) and the Last Glacial Maximum (LGM). The data consists of nine temperature-related variables and eight precipitation-related variables for each period. Specifically, the temperature-related variables are: annual mean temperature; temperature seasonality; maximum temperature of the warmest month; minimum temperature of the coldest month; annual temperature range; mean temperature of the wettest quarter; mean temperature of the driest quarter; mean temperature of the warmest quarter; and the mean temperature of the coldest quarter. The precipitation-related variables are: annual precipitation; precipitation of the wettest month; precipitation of the driest month; precipitation seasonality; precipitation of the wettest quarter; precipitation of the driest quarter; precipitation of the warmest quarter; and precipitation of the coldest quarter. These climate data were retrieved from multiple sources. The modern climate data were obtained from WorldClim (www.worldclim.org), while the paleoclimate data in the MidH and the LGM were retrieved from the Community Climate System Model (CCSM4; www.cesm.ucar.edu/models/ccsm4.0/)[50], with a spatial resolution of 2.5 arc minutes.

Principle component analysis was used for the climate variables in each period to eliminate multicollinearity. The scores in the first principle component, explaining between 70 and 85% of the variance, were used to represent the integrative climate conditions for each epoch.

**Soil property data.** Both physical[29,30] and chemical properties[21,51] of soil may affect the spatial pattern of soil carbon stock. For example, a positive relationship between cation exchange capacity (CEC) and SOCC has been widely reported[52,53]. Therefore, we synthesized data with multiple key soil physical (i.e., soil moisture and texture), and geochemical factors, such as CEC, K, P and pH.

The spatially explicit soil property data were sourced from publicly available databases or were derived from spatial interpolation of site observations as described below. The root-zone soil moisture data were obtained from version 3.0a of the GLEAM data set[54], while soil texture data was taken from the National Earth System Science Data Sharing Infrastructure, National Science & Technology Infrastructure of China (http://www.geodata.cn). Total K and P were extrapolated using Kriging interpolations to cover the missing values. The interpolation analyses were performed using the Geostatistical Toolbox of ArcMap 10.0 (Environmental Systems Research Institute, Inc., Redlands, CA, USA). Soil pH and CEC data were extracted from the SoilGrids data set (http://data.isric.org/geonetwork/srv/chi/catalog.search#/metadata/5333b1af-7620-407f-8bca-2303fc5c7288) with a spatial resolution of 250 m.

The climate (both modern and paleoclimate) and soil properties data were used to assess the relative importance of modern and paleoclimate in affecting modern soil carbon stock using a variety of statistical methods, and then subjected to the SVM model to predict the modern carbon stock over the entire study region. Note that the modern climate and soil property data are based on observation data sets, while paleoclimate data were retrieved from the Community Climate System Model as described above.

**Vegetation and topography data.** Both vegetation type and vegetation coverage were considered. The vegetation type information for each observational site was obtained from Vegetation Atlas of China maps with a scale of 1: 1 000 000[55]. To determine vegetation coverage, the remotely sensed Normalized Difference Vegetation Index (NDVI) was used. NDVI is designed to represent vegetation biomass and subsequent carbon input for soil carbon stock. The NDVI data were derived from the GIMMS NDVI3g data set (https://ecocast.arc.nasa.gov) with a horizontal resolution of 0.083° and a 15-day interval.

In addition to the factors described above, we also included some basic site information such as altitude, slope and relief intensity, as well as geomorphological information, by categorizing the observational sites into the following four landforms: plain; medium-gradient hill; high-gradient hill; and high-gradient mountain. The landform, slope and relief intensity were derived from the Soil and Terrain Database (SOTER) for China (http://data.isric.org/geonetwork).

**Quantifying the relative importance of the paleoclimates.** We used a combined approach involving multiple statistical models to analyse the relative importance of paleo- and modern climate. Specifically, random forest analysis was used to rank the relative importance of the predictors. In this method, the importance of each predictor is determined by evaluating the decrease in prediction accuracy, that is, increase in the node purity, as measured by the decrease in sum of squares between

observations and predictions when it was removed from the model[3]. These analyses were conducted using the RandomForest package in the R statistical software.

Variation partitioning modelling was used to identify the relative importance of paleoclimate and modern climate on soil carbon stock. Variation partitioning is an invaluable tool, as it can identify the individual contributions of a group of predictors of interest and joint contributions between predictors to a given response variable[56]. Hence, it allows an independent portion of the variance to be attributed to the climate variables from mid-Holocene and Last Glacial Maximum periods that cannot be ascribed to the current climate. Variation partitioning analyses were conducted with the R package vegan.

Structural equation modelling (SEM) was used to identify the direct and indirect (via soil properties) effects of paleoclimate on soil carbon stock, and to evaluate the contributions of these factors by assessing the degree of the standardized total effect (direct effect plus indirect effect)[57]. SEM is characterised by its utility for partitioning direct and indirect effects of many predicted variables on the response variable using covariance[58], to help understand complex natural systems. All the SEM analyses were conducted using AMOS 21.0.

In addition, we used partial correlation analysis between the soil carbon stock and the current climate with soil variables controlled separately, and with all soil properties controlled, to identify the effect of the current climate on the soil carbon stock. Changes in the relative importance of paleoclimate with soil depth were investigated using deep soil profiles and the R package relaimpo[59]. We quantified the relative contributions of the regressors using the Lindeman–Merenda–Gold method. This method decomposes $R^2$ into non-negative contributions that automatically sum to the total $R^2$, with bootstrap confidence intervals to assess the stability of the ranking[60].

**Freezing index changes from LGM to MidH**. To understand whether there was intensive permafrost thawing at the mid-Holocene, we calculated the freezing index (FI, Eqs. 2–3) for LGM and MidH periods using paleoclimate simulations from the CCSM model, and analysed the relative change of the freezing index from LGM to MidH to infer the change of permafrost extent. The freezing index was calculated by accumulating the average daily temperatures below 0 °C[61].

$$FI = \sum_{i=1}^{N} |T_i|, \, T_i < 0\,°C \quad (2)$$

$$FI\,change(\%) = (FI_{LGM} - FI_{MidH})/FI_{LGM} * 100 \quad (3)$$

**Prediction and modelled simulations of soil carbon stock**. Machine learning techniques have been proved to be a powerful tool for soil carbon predictions[30,62,63]. Here, we used several different machine learning algorithms to estimate SOCD over the permafrost regions of the Tibetan Plateau. These were: Support Vector Machine (SVM); Random Forest; Artificial Neural Network; Classification and Regression Trees; and Multivariate Adaptive Regression Splines. The results of the leave-one-out cross-validation suggested that SVM showed the best performance in the prediction of the top 30 cm SOCD ($r^2 = 0.65$, $P < 0.001$) (Supplementary Fig. 6).

We, therefore, used SVM and a high resolution (250 m) soil depth data set (http://data.isric.org/geonetwork/srv/chi/catalog.search#/metadata/f36117ea-9be5-4afd-bb7d-7a3e77bf392a) to estimate the top 30 cm soil carbon stock in a spatially explicit manner over the permafrost regions of the Tibetan Plateau. Three-metre soil carbon stock was then derived by extrapolation from the 30 cm soil carbon stock based on the SOCD-soil depth relationship function (Supplementary Fig. 12). Note that for the regions with a soil depth of less than 3 m, the soil carbon stock was adjusted by actual soil depth data. According to the limited existing evidence on the plateau, soils deeper than 3 m in depth may also store a certain amount of soil carbon[23]. However, the small number of deep sample sites on the plateau (only 11 cores are available), and their severely biased spatial distribution, means that it's currently impossible to make a reliable soil carbon estimate for the whole plateau for depths greater than 3 m.

An uncertainty estimate that originated from sampling sites and vertical interpolation of soil carbon stock was provided for the estimation of soil carbon stock on the Tibetan Plateau. To account for the uncertainty introduced by sampling sites, we adopted a bootstrap method (random sampling with replacement) to generate 1000 pseudo replicates, which were used to establish the SVM model in estimating the 30 cm soil carbon stock. Here, we relied on a regression model for vertical extrapolation of deeper-layer soil carbon stock from the top 30 cm layer. The regression model was derived from the relationship between soil organic carbon density and soil depth across 325 sites with deep soil profile data. To estimate the uncertainty due to the vertical extrapolation, we adopted the Monte Carlo sampling technique to draw 1000 random sets of predicted values from their normal distributions with the estimated mean and standard deviations obtained from the regression model for each grid. These two types of uncertainty were merged to yield an estimate of the uncertainty in Tibetan soil carbon stock.

The modelled simulations of soil carbon stock were derived from global gridded (at a 0.5° spatial resolution) outputs (version 1) of 11 terrestrial biosphere models which took part in the Multi-scale Synthesis and Terrestrial Model Intercomparison

Project (MsTMIP)[64]. The models used were: CLASS_CTEM, CLM4, CLM4VIC, DLEM, GTEC, ISAM, LPJ_wsl, ORCHIDEE-LSCE, SiBCASA, TEM6, and VEGAS2.1. All the global model simulations were conducted with similar forcing data, spin-up procedures, and boundary condition[39]. The simulated carbon pools came to the equilibrium after the model spin-up in MsTMIP experimental design (https://nacp.ornl.gov/MsTMIP_variables.shtml). The steady-state criterion for carbon fluxes is that the 100-year mean change in total ecosystem carbon stock must be below 1 g m$^{-2}$ yr$^{-1}$ during the model spin-up[35]. The soil depth layer across models ranges from 1 to 36 m (Supplementary Fig. 13, Supplementary Table 3). We used soil carbon simulations from the SG3 scenario, with time-varying forcing of climate, land use history and atmospheric $CO_2$ concentration[39], for the period 1975-2010, and resampled the output to 0.1º resolution to facilitate comparison with the observation-based predictions.

**Model-observation similarity analysis of soil carbon stock**. We used the Comparison Map Profile (CMP) method[34] to examine spatial similarity between the output of the models and our estimate. Specifically, we calculated the relative distance (RD) and cross correlation (CC) at scales from 1 to 20 (scale 1, 5 and 10 representative of $3 \times 3$ pixel, $11 \times 11$ pixel and $21 \times 21$ pixel moving windows, respectively). The arithmetically averaged values of all mono-scale RD and CC maps were used to examine spatial similarity between the observations and the model simulations (Eqs. 4–6).

$$RD = (\bar{x} - \bar{y})/\bar{y} * 100 \quad (4)$$

$$CC = \frac{1}{N^2} \sum_{i=1}^{N} \sum_{j=1}^{N} \frac{(x_{ij} - \bar{x}) \times (y_{ij} - \bar{y})}{\sigma_x \times \sigma_y} \quad (5)$$

$$\sigma_x^2 = \frac{1}{N^2 - 1} \sum_{i=1}^{N} \sum_{j=1}^{N} (x_{ij} - \bar{x})^2 \quad (6)$$

where $\bar{x}$ and $\bar{y}$ represent averaged values of modelled and observational SOCD over moving windows, respectively; $x_{ij}$ and $y_{ij}$ are the pixel value at row $i$ and column $j$ of the two moving windows for the compared soil carbon stock maps. $\sigma_x$ and $\sigma_y$ are the standard deviations calculated within the two moving windows.

## Data availability

The authors declare that the majority of the data supporting the findings of this study are available through the links given in the paper. The unpublished data are available from the corresponding author upon request.

## Code availability

The new estimate of Tibetan soil carbon stock and R code are available in a persistent repository (https://figshare.com/s/4374f28d880f366eff6d).

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

## Acknowledgements

This study was supported by the Strategic Priority Research Program (A) of the Chinese Academy of Sciences (XDA20050101), the second Tibetan Plateau Scientific Expedition and Research Program (2019QZKK0606), the National Natural Science Foundation of China (41871104, 41530528), and Key Research and Development Programs for Global Change and Adaptation (2017YFA0603604). Jinzhi Ding acknowledges the General (2017M620922) and the Special Grade (2018T110144) of the Financial Grant from the China Postdoctoral Science Foundation.

## Author contributions

T.W., S.L.P. and J.Z.D. conceived the research, J.Z.D. performed statistical analyses, J.Z.D. and T.W. wrote the paper, P.S., S.P.W., Z.J.Y., S.R., J.S.H., Y.W.L., D.L. and G.L.Z. contributed to the writing, L.Z., S.Y. C., F.Q.D., Y.N.L., and Y.H.Y. provided soil carbon data, J.F.M., H.Q.T., X.Y.S., N.Z. and A.A. provided and interpreted the model simulations.

## Additional information

**Competing interests:** The authors declare no competing interests.

