## [Peer Review File · Nature Communications]

Reviewers' comments:

Reviewer #1 (Remarks to the Author):

This manuscript uses an extensive dataset of soil carbon stock measurements across the Tibetan plateau in order to understand how both paleo and modern climate effected the modern-day carbon stocks in that area. They found that modern day precipitation was the single most important factor for determining soil carbon, however, paleo-temperature also played a very important role. They also found that paleo-climate becomes more important in explaining the soil carbon stock distribution at deeper soil depth. The author's found that regional estimates of soil carbon based on the site measurements were nearly double those that were simulated for ecosystem models suggesting that these models need to account for paleo temperature to provide better soil carbon estimates. This work has important applications for improved carbon stock estimates globally, important implications for climate, and can also serve as validation for ecosystem models.

This work could have benefitted from a much more nuanced discussion regarding the cause of the model-observation mismatch in soil carbon. Whereas the authors suggest the underestimation of the ecosystem model soil carbon is from neglecting paleo temperature, a deeper discussion would be useful. From Figure 4 it is clear that the observed soil carbon across these region does overlap with a sub-set of models, and the underestimation of soil carbon is not universal across models. In fact, a small subset of models with extremely low values of SOCD seem to drag the model average down. This observation-model comparison could have benefitted from an uncertainty estimate of the observed soil carbon (authors provide a point estimate but clearly there is uncertainty based upon assumption of vertical profile and limited site data), and also a deeper investigation/discussion into what distinguishes models that are statistically identical to the observations (at least LPJ-WSL, TEM6, ISAMS) and models that barely simulate any soil carbon (CLASS-CTEM, CLM4VIC) Clearly there is much more going on here than simply neglecting paleo-climate (as stated in the conclusions) that causes the range in model output. Spatial maps of the biases and RMSE and R2 between the model and observations could help to better diagnose the differences.

Detailed comments:

Line 40: "paleoclimatic upheavals": strange terminology, perhaps 'transitions' or 'variability' or 'extremes' would be better wording.

Line 41: data from 1114 sites: But roughly, where? Presumably the Tibetan permafrost region.

Line 46: "We derive a new estimate of soil carbon stock". Make this clearer. Presume this means 'modern day' soil carbon stock.

Line 50: "Tibetan and beyond" sounds strange. Perhaps "simulating both Tibetan and global permafrost soil carbon" is better.

Line 58-59: Strange wording: Maybe simplify: ".....where much of the carbon is locked in a frozen state, therefore the soil carbon is only susceptible to change from the most extreme paleoclimate events."

Line 59: Instead of 'upheavals' I think you mean climate 'extremes' here.

Lines 61: Should say: " Land regions of permafrost constitute"

Line 63: Why is Polar Regions capitalized?

Figure 1: Increase the size of the northern hemisphere view of permafrost areas in this figure, but also include the outline of the continents as well. It's strange to just see the permafrost areas by themselves without the continents for reference.

Figure 1: Would be useful to refer to Figure (S6) where deep observation sites are also located. Might be helpful to identify the sites where the deep soil measurements were taken directly in Figure 1, by outlining colored dot in black. Are there significant carbon stores below 3 meters?

Also here and throughout the manuscript it wasn't clear whether the SOCD measurements at 30 CM and at 2 m depth, where point measurements or the average of the entire column (0-30 cm) and (0-2 M) respectively. Could you make that clear here, and throughout the manuscript?

Figure 2: Provide parentheses for the figure caption panel labels.

Line 105: Here you say top 30 cm soil layer, but Figure 1 is at a depth of 30 cm? Are these things different or does Figure 1 show the top 30 cm as well?

Line 112-113: "...where paleo-precipitation was found to be the main driver". Does this mean overall, or for just paleo climate?

Line 548: include a comma here: when none of, each of, and

Line 129: Pedogenesis : define or use another word.

Figure S2: Not clear what the numbers mean for relative importance of model variables. Assume the lower coefficients values which are listed first are most important?

Table S1: SOCD at depth of 30 cm. So here you are looking at SOCD at a specific depth, but in the text you are talking about the top 30 cm. Should you be switching back and forth?

Line 136-144: A bit confusing: "the direct effect, however, should not be overstated." But isn't that consistent with your findings, and Figure 2c where the indirect effects from the LGM are more important than the direct effects? So, I am not sure why you think the direct effects were 'overstated'. Instead focus this paragraph on the physical explanation of why the indirect effects are more important than the direct effects from the LGM --- because the Mid Holocene period obscured the direct effects.

Figure 3: Should state in caption that the relative important was determined through the LMG method for clarity.

Line 169-173: This is an awkward sentence, is hard to understand, and should be re-written. I think you mean something like: "We should caveat our findings regarding the impact of paleoclimate on soil carbon distribution with depth, because the paleoclimate parameters are model-derived, and not validated from proxy data."

Also does this caveat apply to your results (Figure 2) or just to the LMG method in Figure 3?

Figure S3: Should say in caption that these factors were used for machine learning model algorithms to estimate soil C.

Figure S4: Should state in caption that in panel (b) is 1:1 is only SVM model because it provide the best predictions. Also is this for top 30 cm depth or just for 30 cm depth? Just be clear here and throughout.

Lines 197-203, Figure 4: Unclear what assumptions each model makes about soil layer depth, and what this means for total carbon stock. Not really explained in the text.

Figure 4: Showing a Taylor Diagram based on spatial correlation between observed and modeled carbon stocks is helpful, but wouldn't spatial maps of carbon stock biases, and spatial maps of RMSE be much more informative in revealing where and how the models fail?

Line 217-219: Perhaps lack of inclusion of paleo-climate temperature could have led to some of the biases for ecosystem models, but what about discussing some of the other mechanistic assumptions with the models themselves or the soil characteristic maps that go into them? For example turnover time of soil carbon pools etc.? What is different between LPJ-WSL and TEM6 such that they overestimate the SOCD, where other models are much below? The distribution of models do encapsulate the observed SOCD so perhaps this can be investigated further.

Also are all models simulating the same soil layer depth such that SOCD can be used universally across models— such as what you have been doing through the entire manuscript?

Line 230-232: It seems odd to mention SOCD variation with depth here in the conclusions and contrast that to non-permafrost regions, when this was not mentioned in the main paper --- at least was not a focus.

Line 236: "Our new estimate of the carbon pool, obtained by including paleoclimate as an additional predictor, is double the size of current modelled values. Future modelling of soil carbon cycling should include paleoclimate information during the model spin-up period so as to accurately represent the impacts of paleoclimate on soil properties."

So this seems like a major leap in conclusions. First off, how confident are you in the estimate of your Tibetan soil carbon stock value and SOCD? Clearly there is a limited number of sites, and a limited number of deep sites, so do you have an estimate of uncertainty based upon your site measurement approach (Figure 4)? It is likely that the uncertainty in the carbon stocks overlaps and is statistically indistinguishable from at least 2-3 models and likely as many as 5 of the ecosystem models. Looking at Figure 4, it looks like the SOCD average is dragged down by just a handful of

models which simulate barely any SOCD at all including CLASS-CTEM, CLM4VIC and SIBCASA. What is going on with those models? I think a more nuanced discussion of uncertainty and also some mechanistic explanation in why there is so much variation between ecosystem models, would be more insightful.

Also are paleo-climate data even available in model format? Precipitation and temperature are likely available but models require things like relative humidity, long and shortwave radiation etc. Models also function on sub-daily timesteps so that there needs to be a way to downscale coarse paleo climate data to something that is model-ready.

Reviewer #2 (Remarks to the Author):

The authors apply statistical analysis methods to analyse the relationship between permafrost distribution (and various soil properties) data collected in the field, and climate indicators derived from data (present-day) and model outputs (mid Holocene and LGM). They also provide a new estimate for total soil carbon in the Tibetan permafrost region and compare this to recent modelling efforts.

A main finding is that modelling the permafrost region of Tibet using only present-day climate conditions results in an underestimate of total carbon in those soils (and that this conclusion may be extended to other permafrost regions).

Overall this is an interesting and important study, quantifying the effect of neglecting paleoclimate forcing in the modelling of permafrost carbon stock. It makes extensive use of correlating large datasets, and it looks like a lot of work has gone into the study. It relies heavily on correlation statistics, but this is probably an appropriate method, and those results should certainly be reproducible with the datasets.

It would nice to have some more discussion on the physical mechanisms that result in paleo climate imprint on existing permafrost soils, with a schematic in the style of figure s3, to clarify the argument.

I have just a few comments to the authors on some specific items:

Line 177: It would be useful to clarify what is the indirect mechanism by which soil properties affect organic carbon stabilisation (is this chemical, mechanical ?)

Line 133: You say the 14C indicates that the *distribution* of permafrost is largely a relic of that formed 35ka to 10.8ka BP. Does line 133 to 135 then suggest very little re-advance of permafrost since the mid Holocene?

Line 140: This paragraph states that during the mid-Holocene there was probably a deepening of the active layer, and this assumption is used to support a conclusion that soil properties are probably the controller of paleoclimate legacies on soil carbon stocks. I think this point needs to be expanded. Is it reasonable to assume that the active layer deepened a lot at the mid-Holocene? How would this manifest in your data? Frost index has been used as an indicator for permafrost extent, and that is sensitive to seasonal insolation. 6ka was in a falling Northern hemisphere summer insolation period, affecting the freezing and thawing cycle, how does the temperature seasonality variable change since the NH summer insolation maximum (~11ka) to the mid-Holocene?. Perhaps there was not a *large* top-down deepening of the active layer, and how would this affect your interpretation?

Line 144: Is the top 10cm of soil actually representative of the top 30cm (which is the basis of correlations in figure 2)? If the same accumulation rates hold until 30cm (which is probably not the case), these soils would be older than mid-Holocene age. What do you consider to be “paleo-vegetation”? I assume you mean LGM-age vegetation. Why do you not have 14C dating up to 30cm (at least in some places)? Is it possible that actually the soils at 30cm are already LGM age?

Line 190: There does not appear to be an uncertainty value on the total estimated carbon stock for Tibet permafrost derived in this study, which would be useful.

Reviewer #3 (Remarks to the Author):

The paper describes an innovative and novel approach to study climate-driven past permafrost soil processes and consequent inherited legacies, which led to formation of current carbon stock in Tibetan mountains. The authors conclude that representative palaeoclimate data combined with information of soil physicochemical properties yields more reliable carbon stock estimations, than if

the palaeoclimate reconstruction data is not properly applied, i.e. when ecosystem modelling only accounts for the pre-industrial climate conditions. The new carbon stock estimate doubled the size of the C storage.

Basically, I have only positive things to say. The paper is very well written and the study description proceeds logically. The interest is not lost, despite various steps are taken and a huge set of tools and study components is described and used. It is clear that the authors have used lot of time and energy in trying identifying the most suitable ways to understand relationship between (palaeo)climate and past and current (and future) carbon dynamics. The study is also exploiting extensive field data collected over the years. I am very happy to see palaeoclimate aspect highlighted in understanding current and future C processes. Undoubtedly, extrapolations and generalisations used in this study infold uncertainties and some assumptions, for instance soil depths and stored C, have to be considered tentative. However, I felt that the authors were aware of the limitations. My background is not in modeling but I found it very interesting to read how various models were step-by-step applied to quantify different variables and in-between linkages. A respectable amount of different data-bases were exploited. The study is local/regional and as such does not have global implications. (I leave it to the editors to decide if the novelty of study method is enough to allow publication in Nature Communications, or would a more traditional palaeo journal, such as QSR, be a better forum). In any case, as the carbon stock estimate quite radically changed for Tibetan permafrost area when palaeoclimate was better accounted, the approach should be further applied to other regions from where C stock model estimations exist. If similar increase in stock values is repeated, larger-scale model revisions are needed.

One annoying detail: "megathermal", established term is Holocene Thermal Maximum HTM

Response to Reviewer #1

[Comment 1] This manuscript uses an extensive dataset of soil carbon stock measurements across the Tibetan plateau in order to understand how both paleo and modern climate effected the modern-day carbon stocks in that area. They found that modern day precipitation was the single most important factor for determining soil carbon; however, paleo-temperature also played a very important role. They also found that paleo-climate becomes more important in explaining the soil carbon stock distribution at deeper soil depth. The author's found that regional estimates of soil carbon based on the site measurements were nearly double those that were simulated for ecosystem models suggesting that these models need to account for paleo temperature to provide better soil carbon estimates. This work has important applications for improved carbon stock estimates globally, important implications for climate, and can also serve as validation for ecosystem models.

[Response] Thanks for the reviewer's insightful and positive comments. The detailed comments listed below enabled us to revisit our findings, and conduct a thorough revision on the original manuscript. After incorporating these comments, we believe that the revised manuscript has been significantly improved.

[Comment 2] This work could have benefitted from a much more nuanced discussion regarding the cause of the model-observation mismatch in soil carbon. Whereas the authors suggest the underestimation of the ecosystem model soil carbon is from neglecting paleo temperature, a deeper discussion would be useful. From Figure 4 it is clear that the observed soil carbon across these regions does overlap with a sub-set of models, and the underestimation of soil carbon is not universal across models. In fact, a small subset of models with extremely low values of SOCD seems to drag the model average down. This observation-model comparison could have benefitted from an uncertainty estimate of the observed soil carbon (authors provide a point estimate but clearly there is uncertainty based upon assumption of vertical profile and limited site data), and also a deeper investigation/discussion into what distinguishes models that are statistically identical to the observations (at least LPJ-WSL, TEM6, ISAMS) and models that barely simulate any soil

carbon (CLASS-CTEM, CLM4VIC). Clearly there is much more going on here than simply neglecting paleo-climate (as stated in the conclusions) that causes the range in model output. Spatial maps of the biases and RMSE and R2 between the model and observations could help to better diagnose the differences.

[Response] Thanks for your constructive comments. Following your suggestion, we have provided an uncertainty estimate of observed soil carbon stock, and then compared it with weighted model simulations using the Bayesian Model Averaging method¹. In addition, a spatially explicit assessment of model-observation mismatch using the Comparison Map Profile method² is also provided.

First, an uncertainty estimate that originated from sampling sites and vertical interpolation of the soil carbon stock was provided for the estimation of soil carbon stock on the Tibetan Plateau. To account for the uncertainty introduced by sampling sites, we adopted a bootstrap method (random sampling with replacement) to generate 1,000 pseudo replicates, which were used to establish the SVM model in estimating the 30 cm soil carbon stock. Here we relied on a regression model for vertical extrapolation of deeper-layer soil carbon stock from the top 30 cm layer. The regression model was derived from the relationship between soil organic carbon density and soil depth across 325 sites with deep soil profile data. To estimate the uncertainty due to the vertical extrapolation, we adopted the Monte Carlo sampling technique to draw 1,000 random sets of predicted values from their normal distributions with the estimated mean and standard deviations obtained from the regression model for each grid. These two types of uncertainty were merged to yield an estimate of the uncertainty in Tibetan soil carbon stock. Our analysis showed that the Tibetan soil organic carbon pool to a depth of 3 m was 36.6 PgC (95% confidence range: 34.2 - 38.9 PgC), and the mean soil organic carbon density was estimated to be 15.4 kg C m⁻² (95% confidence range: 14.4-16.4 kg C m⁻²). To fully resolve your concerns, we have added the following text into the Methods section: “*An uncertainty estimate that originated from sampling sites and vertical interpolation of soil carbon stock was provided for the estimation of soil carbon stock on the Tibetan Plateau. To account for the uncertainty introduced by sampling sites, we adopted a bootstrap method (random sampling with replacement) to*

generate 1,000 pseudo replicates, which were used to establish the SVM model in estimating the 30 cm soil carbon stock. Here we relied on a regression model for vertical extrapolation of deeper-layer soil carbon stock from the top 30 cm layer. The regression model was derived from the relationship between soil organic carbon density and soil depth across 325 sites with deep soil profile data. To estimate the uncertainty due to the vertical extrapolation, we adopted the Monte Carlo sampling technique to draw 1,000 random sets of predicted values from their normal distributions with the estimated mean and standard deviations obtained from the regression model for each grid. These two types of uncertainty were merged to yield an estimate of the uncertainty in Tibetan soil carbon stock.” (Lines 479-492 on Page 21-22).

In addition, the numbers related to the Tibetan soil carbon stock and soil carbon density were also updated accordingly throughout the manuscript.

Second, we compared our estimate of Tibetan soil carbon stock with the outputs of the terrestrial ecosystem models, and found that neither the total amount nor the spatial distribution of soil carbon stock were reliably simulated by these models. Consistent with our previous findings, the soil carbon stock has generally been underestimated by the ecosystem models except TEM6 and LPJ-wsl (**Figure R1a**). The spatial correlation coefficients between the modelled values and our empirical estimates across grid cells were overall less than 0.5, indicating variations in the spatial distribution of soil carbon stock are not well represented by the ecosystem models (**Figure R1b**).

Furthermore, we used the Comparison Map Profile (CMP) method² to examine the spatial similarity between model output and observations. Specifically, we calculated the relative distance (*RD*) and cross correlation (*CC*) at scales from 1 to 20 (scale 1, 5 and 10 representative of 3×3 pixel, 11×11 pixel and 21×21 pixel moving windows, respectively). The arithmetically averaged values of all mono-scale *RD* and *CC* maps were used to examine spatial similarity between observations and model simulations.

$$RD = (\bar{x} - \bar{y}) / \bar{x} * 100 \quad (1)$$

$$CC = \frac{1}{N^2} \sum_{i=1}^N \sum_{j=1}^N \frac{(x_{ij} - \bar{x}) \times (y_{ij} - \bar{y})}{\sigma_x \times \sigma_y} \quad (2)$$

$$\sigma_x^2 = \frac{1}{N^2-1} \sum_{i=1}^N \sum_{j=1}^N (x_{ij} - \bar{x})^2 \quad (3)$$

Where \bar{x} and \bar{y} represent averaged values of modelled and observational SOCD over moving windows, respectively; x_{ij} and y_{ij} are the pixel value at row i and column j of the two moving windows for the compared soil carbon stock maps. σ_x and σ_y are the standard deviations calculated within the two moving windows. The related text for the CMP method has been added into the Method section of the updated manuscript (lines 503-517, pages 22-23).

We found that eight out of eleven models, particularly CLASS_TEM, CLAM4VIC, GTEC and SiBCASA, clearly underestimated the Tibetan soil carbon stock (**Figure R2**). Even the models such as LPJ-WSL and TEM6, which simulated a total soil carbon stock comparable to the observations, still failed to capture the spatial distribution of soil carbon stock, as indicated by the low degree of spatial agreement (low correlation values) (**Fig. R1b**; **Fig R3**). For these two models, the overestimation in the western Tibetan Plateau effectively compensates for the underestimation in the east, leading to a model-data match in terms of the total soil carbon stock (**Fig. R2**). We have added the following text in the Results section in the updated manuscript as a spatially explicit assessment of the model-data mismatch. “*We further calculated the spatially explicit indices relative distance (RD) and cross correlation (CC) at multiple scales to examine spatial similarities between the models and our estimate using the Comparison Map Profile (CMP) method². We found that eight out of eleven models, particularly CLASS_TEM, CLAM4VIC, GTEC and SiBCASA, clearly underestimated the Tibetan soil carbon stock (Supplementary Fig. 8). We also observed a large spatial inconsistency between simulated soil carbon stock and our estimates, with a low correlation in most parts of the study area (Supplementary Fig. 9). Even the models such as LPJ-WSL and TEM6, which simulated a total soil carbon stock of comparable size to our estimate, still fail to capture the spatial distribution of soil carbon stock, as indicated by the generally low values of correlation (Supplementary Fig. 9). For these two models, the overestimation in the western Tibetan Plateau effectively compensates for the underestimation in the east, leading to an apparent model-observation match in terms of the total soil carbon stock*

(Supplementary Fig. 8)” (lines 220-233, page 10-11).

As stated by the reviewer, there is a substantial variation in the models’ performance in simulating Tibetan soil carbon stock. For example, the models LPJ-WSL and TEM6 simulated the size of the soil carbon stock to be statistically close to our estimate, but other models such as CLASS-CTEM and CLM4VIC showed considerable underestimation (**Figure R1**). The arithmetic average of modeled soil carbon stock may be dragged down by extremely low output values from models that suffer from deficiencies in the representation of the main processes affecting soil carbon stock, such as the vegetation carbon input (e.g. NPP) and soil organic carbon turnover^{3, 4}. **We thus applied the Bayesian Model Averaging method (BMA), which is conditional on an independent observation data¹, to tone down the role of models that have notable deficiencies in the simulation of major physiological processes.** Here we selected net primary productivity (NPP), an indicator of vegetation carbon input, to rank the model performance given the availability of satellite-derived NPP product⁵. The larger weights were assigned to models that have better performance in simulating NPP with respect to satellite-derived observations. Our results showed that the weighted ensemble mean (11.5 ± 4.2 s.e.m PgC) was even lower than the arithmetic model average (16.9 ± 4.1 PgC), confirming the significant underestimation of Tibetan soil carbon stock by these ecosystem models. It thus suggested that model underestimation is not due to a few extreme low simulations, but rather a general problem faced by ecosystem models in simulating permafrost carbon stock on the Tibetan Plateau. To resolve your concern, we have added the following text into the revised manuscript “*Here we used the Bayesian Model Averaging method (BMA), which is conditional on an independent observation data¹, to tone down the role of models that have notable deficiencies in representing major physiological processes. Given the availability of satellite-derived NPP product⁵, we adopted NPP, as an indicator of vegetation carbon input, to rank the model performance. Larger weights were assigned to models that have a better performance in simulating NPP with respect to satellite-derived observations.*” (lines 208-214, page 10). The weighted ensemble mean of the model outputs was used to replace the original arithmetic model average in the revised manuscript (lines 214-216, page 10).

To demonstrate the importance of integrating the paleoclimate data in model spin-up, we calculated the difference between the ensemble mean of model simulations (paleoclimate not considered) and our estimation (paleoclimate considered), and then correlated to the paleoclimate. We found that the lack of paleoclimate is significantly correlated to the model-observation discrepancies. Additionally, we found a nearly perfect relationship between the initial soil carbon stock after model spin-up and present-day (1980–2010) stock across models ($r^2 = 0.99$, $P < 0.001$; **Figure R4**), highlighting a critical role of integrating the paleoclimate data in model spin-up. We have added the following text to emphasize the importance of integrating the paleoclimate data in model spin-up “*There was a nearly perfect relationship between the initial soil carbon stock after model spin-up and the present-day (1980–2010) stock across the models ($r^2 = 0.99$, $P < 0.001$; Supplementary Fig. 10), highlighting the critical role of model spin-up in the estimation of soil carbon stock*” (lines 234-237, page 11).

Besides the neglect of paleoclimate in model spin-up, we strongly agree with the reviewer that other processes that were not well resolved in current ecosystem models, such as soil carbon turnover time, could also account for the model underestimation. **First**, poor model representation of long turnover time of deep soil carbon, especially in permafrost-affected regions, may lead to significant underestimation of soil carbon stock^{6, 7}. The current soil carbon models adopted a single vertically integrated soil carbon pool, without considering the vertical gradients in soil carbon stability and decomposability⁶. Actually, part of the organic carbon stored in deep layers, many thousands of years older than the surface organic carbon, is generally considered to be stable due to low decomposition rates, especially in permafrost-affected regions^{8, 9}. The lack of this vertical dimension in modeling soil carbon cycling may thus contribute to the model underestimation. This statement was in line with our model-observation comparison analysis, which showed that there was a higher degree of underestimation occurring in permafrost-affected soils than in non-permafrost affected soils (**Figure R5**). **Second**, several typical alpine vegetation types such as marsh meadow and alpine meadow, which were characterized by relatively high organic carbon density and slow soil carbon turnover rate, were not well represented in the models of the MsTMIP protocol¹⁰.

To test if this limitation contributes to the model underestimation, we compared the relative distance of observation and model among different vegetation types, and found that soil carbon stock was severely underestimated in marsh meadow (mean relative distance = -88%) and alpine meadow (mean relative distance = -64%) (**Figure R5**). These two types of alpine vegetation cover about one third of the total area of the plateau, accounting for 41% of the total soil carbon stock on the plateau, and thus represent an important contribution to the model underestimation. In order to fully resolve the reviewer's concern, we have added the following paragraph into the revised MS: *“In addition to paleoclimates, other processes that are not well resolved in current ecosystem models, such as soil carbon turnover time, may also account for some of the model underestimation. Firstly, poor model representation of the long turnover time of deep soil carbon, especially in permafrost-affected regions, may lead to significant underestimation of soil carbon stock^{6, 7}. The current soil carbon models adopt a single vertically integrated soil carbon pool, without considering the vertical gradients in soil carbon stability and decomposability⁶. In reality, the part of the organic carbon stored in deep layers, many thousands of years older than the surface organic carbon, is generally considered to be stable due to low decomposition rates, especially in permafrost-affected regions^{8, 9}. Therefore, it's possible that the omission of this vertical dimension in the modelling of soil carbon cycling may contribute to the models' underestimation. This hypothesis is in agreement with our model-observation comparison analysis, which showed that there was a higher degree of underestimation in permafrost-affected soils than in non-permafrost affected soils (Supplementary Fig. 11). Secondly, several typical alpine vegetation types, such as marsh meadow and alpine meadow, which are characterized by relatively high organic carbon density and slow soil carbon turnover rates, were not well represented in the models of the MsTMIP protocol¹⁰. To test whether this limitation contributes to the model underestimation, we compared the relative distance between the observed and modelled soil carbon stock for different vegetation types. We found that the soil carbon stock was severely underestimated in marsh meadow (mean relative distance = -88%) and alpine meadow (mean relative distance = -64%) (Supplementary Fig. 11). Since these two types of alpine vegetation cover about one third of the total area of the plateau, accounting for 41% of the total soil carbon stock, these underestimates represent a major contribution to the overall model*

underestimation” (Lines 257-281 on Page 12-13).

Figure R1 Comparison of soil C stock simulated by 11 ecosystem models with estimates from this study. **a** shows total soil organic carbon stock. **b** is the Taylor diagram which shows correlation coefficients between the gridded model simulations and estimates from this study, and the normalized standard deviation of model simulations by observations (Obs).

Figure R2 Spatially explicit multi-scale averaged relative distance between model simulations of soil carbon stock and our estimate, using the Comparison Map Profile method².

Figure R3 Spatially explicit multi-scale averaged cross-correlation between model simulations of soil carbon stock and our estimate, using the Comparison Map Profile method².

Figure R4 Dependence of current soil organic carbon stocks simulations on simulation initials in the 11 ecosystem models.

Figure R5 Vegetation-specific bias as indicated by the relative distance of weighted ensemble mean of the 11 model outputs as compared to the new observational estimate in this study. The black dots and lines inside the boxes represent mean and median values, and box ends represent 25th to 75th quartile range, and whiskers denote 10th to 90th quartile range.

[Comment 3] Line 40: “paleoclimatic upheavals”: strange terminology, perhaps ‘transitions’ or ‘variability’ or ‘extremes’ would be better wording.

[Response] Following your suggestion, we have changed this sentence as follows

“Quantifying the impacts of paleoclimatic extremes on soil carbon stock can shed light on the

vulnerability of permafrost carbon in the future” (line 43-45, page 3).

[Comment 4] Line 41: *data from 1114 sites: But roughly, where? Presumably the Tibetan permafrost region.*

[Response] Following your suggestion, we have revised this sentence as follows: *“Here, we synthesized data from 1114 sites across the Tibetan permafrost region to report that paleoclimate is more important than modern climate in shaping current permafrost carbon distribution” (line 45-47, page 3).*

[Comment 5] Line 46: *“We derive a new estimate of soil carbon stock”. Make this clearer. Presume this means ‘modern day’ soil carbon stock.*

[Response] To make it readable, we have revised this sentence as suggested: *“We derive a new estimate of modern soil carbon stock to 3 m depth by including the paleoclimate effects...” (line 49-50, page 3).*

[Comment 6] Line 50: *“Tibetan and beyond” sounds strange. Perhaps “simulating both Tibetan and global permafrost soil carbon” is better.*

[Response] Following your suggestion, we have updated this sentence as follows: *“The discrepancy highlights the urgent need to incorporate paleoclimate information into model initialization for simulating permafrost soil carbon stocks” (line 52-54 Page 3).*

[Comment 7] Line 58-59: *Strange wording: Maybe simplify: “.....where much of the carbon is locked in a frozen state, therefore the soil carbon is only susceptible to change from the most extreme paleoclimate events.”*

[Response] Following your suggestion, we have updated this sentence as follows: *“Such paleoclimate signals would be expected to be strongest in permafrost soils^{9, 11, 12}, where much of the soil carbon is locked in a frozen state, and therefore only susceptible to change from the most extreme paleoclimate events. Only the warmest and coldest periods are likely to leave recognizable changes on the soil carbon in these soils” (line 60-64, page 4).*

[Comment 8] Line 59: *Instead of ‘upheavals’ I think you mean climate ‘extremes’ here.*

[Response] Following your suggestion, we have changed “*upheavals*” into “*extreme*” in the updated manuscript (line 62, page 4).

[Comment 9] Lines 61: Should say: “Land regions of permafrost constitute”...

[Response] Following your suggestion, we have updated this sentence as suggested “*Land regions of permafrost constitute the largest soil carbon pool in terrestrial ecosystems...*” (line 65, page 4).

[Comment 10] Line 63: Why is Polar Regions capitalized?

[Response] We have changed the “*Polar Regions*” into “*polar regions*” to avoid potential misunderstandings in revised manuscript (line 67, page 4).

[Comment 11] Figure 1: Increase the size of the northern hemisphere view of permafrost areas in this figure, but also include the outline of the continents as well. It's strange to just see the permafrost areas by themselves without the continents for reference.

Figure 1: Would be useful to refer to Figure (S6) where deep observation sites are also located. Might be helpful to identify the sites where the deep soil measurements were taken directly in Figure 1, by outlining colored dot in black. Are there significant carbon stores below 3 meters?

Also here and throughout the manuscript it wasn't clear whether the SOCD measurements at 30 CM and at 2 m depth, were point measurements or the average of the entire column (0-30 cm) and (0-2 M) respectively. Could you make that clear here, and throughout the manuscript?

[Response] Following your suggestions, we have revised Figure 1 by increasing the size of the insert which shows the northern hemisphere view of permafrost areas with the continents for reference. We have also indicated the locations of deep sites on the map by using black outlines for the symbols (**Figure R6**).

We provided a clear explanation on the nominations of $\text{SOCD}_{30\text{ cm}}$ and $\text{SOCD}_{200\text{ cm}}$ here and throughout the manuscript: $\text{SOCD}_{0-30\text{ cm}}$ and $\text{SOCD}_{0-200\text{ cm}}$ denote the total soil organic carbon density (SOCD) for the 0~30 cm and the 0~200 cm soil layers, respectively.

According to limited existing evidence on the plateau, soils deeper than 3 m in depth

could also store a certain amount of soil carbon (**Figure R7b**, data from Mu *et al.*, 2015). But the sample sites on the plateau are too few (only 11 cores available) and severely biased in spatial distribution (**Fig R7a**), which precludes a reliable soil C estimation deeper than 3 m over the whole plateau in this study. To express this point, we added some descriptions in the revised manuscript as follows: “*According to the limited existing evidence on the plateau, soils deeper than 3 m in depth may also store a certain amount of soil carbon*¹³. However, the small number of deep sample sites on the plateau (only 11 cores are available), and their severely biased spatial distribution, means that it’s currently impossible to make a reliable soil carbon estimate for the whole plateau for depths greater than 3 m” (lines 473-478, page 21).

Figure R6 The locations and soil organic carbon density (SOCD) in the top 30 cm layer for the 1114 sampling sites over the permafrost regions on the Tibetan Plateau. The deep soil carbon measurements (more than 2 m depth) are indicated by black outlines for the colored dots.

Figure R7 Location map of 11 deep cores from Mu *et al.* (2015) and the vertical distribution of soil organic carbon concentration with soil depth for the deep cores.

[Comment 12] Figure 2: Provide parentheses for the figure caption panel labels.

[Response] Thanks for your suggestion. According to the format requirements of *Nature Communications*, no parenthesis for the figure panel was needed. We thus kept this format without parentheses.

[Comment 13] Line 105: Here you say top 30 cm soil layer, but Figure 1 is at a depth of 30 cm? Are these things different or does Figure 1 show the top 30 cm as well?

[Response] Thanks for pointing out this mistake. It is indeed the top 30 cm soil layer. We are sorry for the confusion caused by this mistake, and have corrected this mistake throughout the manuscript.

[Comment 14] Line 112-113: "...where paleo-precipitation was found to be the main driver". Does this mean overall, or for just paleo climate?

[Response] According to Delgado-Baquerizo *et al.* (2017), paleo-precipitation was the main driver affecting the current soil carbon stock among all investigated factors, including both paleo and modern climates. To avoid potential confusion, we have revised this sentence as follows: "*This finding, however, differs from the results for arid and semiarid regions, where*

paleo-precipitation was found to be the main driver among all the investigated factors, including both paleo and modern climates” (line 116-118, page 6)

[Comment 15] Line 548: include a comma here: when none of, each of, and

[Response] Done as suggested.

[Comment 16] Line 129: Pedogenesis: define or use another word.

[Response] To be specific and clear, we have replaced “Pedogenesis” by “a diversity of soil evolution processes” in the updated manuscript (line 141, page 7).

[Comment 17] Figure S2: Not clear what the numbers mean for relative importance of model variables. Assume the lower coefficients values which are listed first are most important?

[Response] The numbers listed in the legend provide an important diagnosis for the overall performance of the SEM. The specification of an appropriate SEM model required that the difference between the SEM simulation and the observations is not statistically significant ($P > 0.05$)^{14, 15}. The model parameter optimization was then pursued further to decrease values of χ^2 , RMSEA, and AIC and to increase the P value^{14, 15}. We have added more descriptions in the legend of Supporting Information as follows “*The specification of an appropriate SEM model required that the difference between the SEM simulation and the observations is not statistically significant ($P > 0.05$). Model parameter optimization was pursued further by decreasing values of χ^2 , RMSEA, and AIC and to increase the P value so as to improve the model fit to the observations^{14, 15}. The numbers used for SEM are standardized path coefficients, which can reflect the relative importance of the predicted variables within the model. * denotes statistically significant at the 0.05 probability level”.*

[Comment 18] Table S1: SOCD at depth of 30 cm. So here you are looking at SOCD at a specific depth, but in the text you are talking about the top 30 cm. Should you be switching back and forth?

[Response] Sorry for this confusion. It is indeed the top 30 cm soil layer. We have clarified this throughout the manuscript and supplementary material.

[Comment 19] Line 136-144: A bit confusing: “the direct effect, however, should not be overstated.” But isn’t that consistent with your findings, and Figure 2c where the indirect effects from the LGM are more important than the direct effects? So, I am not sure why you think the direct effects were ‘overstated’. Instead focus this paragraph on the physical explanation of why the indirect effects are more important than the direct effects from the LGM --- because the Mid Holocene period obscured the direct effects.

[Response] We strongly agree with the reviewer’s comment that the focus of this paragraph is to explain why the indirect effects of paleoclimate are more important than the direct effects on the soil carbon stock. We thus highlighted that the direct effect of paleoclimate operated at a secondary importance instead of “was overstated” in the revised manuscript. To resolve your concern, we have changed “*The direct effect, however, should not be overstated. If the preservation of paleo-vegetation signals in the upper soil layers were to be the main explanation of the current soil carbon distribution, legacy impacts of the LGM should be significantly concealed by those of the MidH*” into “*...the influence of paleoclimate on Tibetan permafrost soil carbon distribution operated primarily through modifying soil physiochemical properties, with the direct effect taking a secondary role. If the preservation of paleo-vegetation signals in the upper soil layers predominantly explained the current soil carbon distribution, the legacy impacts of the LGM should be significantly concealed by those of the MidH*” (line 145-149, page 7).

[Comment 20] Figure 3: Should state in caption that the relative important was determined through the LMG method for clarity.

[Response] As suggested, we have added a sentence as follows: “*The relative importance was determined by the Lindeman-Merenda-Gold method*” (line 729-730, page 33).

[Comment 21] Line 169-173: This is an awkward sentence, is hard to understand, and should be re-written. I think you mean something like: “We should caveat our findings regarding the impact of paleoclimate on soil carbon distribution with depth, because the paleoclimate parameters are model-derived, and not validated from proxy data.” Also does this caveat apply to your results (Figure 2) or just to the LMG method in Figure 3?

[Response] The paleoclimate data used in this study are model-derived, and need further validation from proxy data over the Tibetan Plateau in the future. This caveat would also apply to parts of Figure 2 where such model-derived paleoclimatic data involved. We thus proposed that future studies may benefit from integration of paleoclimate data with multiproxy data.

Following your suggestion, we have revised the sentence as “*However, we should caveat our findings regarding the impact of paleoclimate on soil carbon distribution, because the paleoclimate parameters are model-derived and not validated from proxy data*” (line 303-305, page 34).

[Comment 22] Figure S3: Should say in caption that these factors were used for machine learning model algorithms to estimate soil C.

[Response] Following your suggestion, we have revised the caption of Figure S3 as follows “*Diagram of factors controlling soil carbon patterns, which were used for machine learning model algorithms to estimate soil carbon stock over Tibetan permafrost regions*”.

[Comment 23] Figure S4: Should state in caption that in panel (b) is 1:1 is only SVM model because it provide the best predictions. Also is this for top 30 cm depth or just for 30 cm depth? Just be clear here and throughout.

[Response] Following your kind suggestion, we have added more clarifications as follows “... *(b) Scatter plot of the observed and the predicted SOCD for Support Vector Machine (SVM) model only because of its better performance in predicting SOCD than other approaches. The dashed line in (b) is the 1:1 line. ...*”. We also clearly described that $\text{SOCD}_{0-30\text{ cm}}$ refers to the sum of soil organic carbon density (SOCD) in the top 30 cm layer throughout the manuscript.

[Comment 24] Lines 197-203, Figure 4: Unclear what assumptions each model makes about soil layer depth, and what this means for total carbon stock. Not really explained in the text.

[Response] The soil depth tends to vary across models^{10, 16}. But for models such as VEGAS and GTEC, there is no actual information on the soil layer depth (**Table R1**). It appeared that the soil depth should not be a major source of the wide range in the simulation of soil carbon

stock, since there is no significant correlation between simulated soil carbon stock and soil depth across models ($r^2 = 0.07$, $P = 0.25$; **Figure R8**). For instance, the soil C stock derived from the CLASS-CTEM model for 4.1 m-depth (1.06 Pg C) is significantly lower than that of LPJ-WSL 1.5 m-depth outputs (37.03 Pg C) (**Table R1**).

To resolve your concern, we provided more information on the soil depth for each model in the supplementary information as Supplementary Table 3, as well as following text in the Supplementary Discussion section: *“In addition, the soil layer depth tends to vary across models (Supplementary Table 3), but it appears that the soil depth is not a major source of the model spread in the simulation of soil carbon stock, since there was no significant correlation between the simulated soil carbon stock and soil depth across 11 models ($r^2 = 0.07$, $P = 0.25$; Supplementary Fig. 13)”*.

Table R1 Comparisons of soil depth and simulations of soil organic carbon stock among the ecosystem models. NA indicates that the soil depth is not available.

Model	Soil depth (m)	Soil organic carbon stock (Pg C)
CLASS-CTEM	4.1	1.06
CLM4	3.82	14.55
CLM4VIC	3.82	0.01
DLEM	1	22.09
GTEC	NA	10.81
ISAMS	3.5	20.72
LPJ-WSL	1.5	37.03
ORCHIDEE	2	14.59
SiBCASA	15	3.92
TEM6	36	41.90
VEGAS2.1	NA	22.60

Figure R8 Scatter plot of the soil carbon stocks on the Tibetan Plateau and the soil depths considered in the 11 ecosystem models.

[Comment 25] Figure 4: Showing a Taylor Diagram based on spatial correlation between observed and modeled carbon stocks is helpful, but wouldn't spatial maps of carbon stock biases, and spatial maps of RMSE be much more informative in revealing where and how the models fail?

[Response] We strongly agree with the reviewer that a spatially explicit diagnosis of the mismatch between modeled SOCD and our estimate could help to better assess the model fit. Here we used the Comparison Map Profile (CMP) method² to examine the spatial similarity between model outputs and observations. Specifically, we calculated the relative distance (*RD*) and cross correlation (*CC*) at scales from 1 to 20 (scale 1, 5 and 10 are representative of 3×3 pixel, 11×11 pixel and 21×21 pixel moving windows, respectively). The arithmetically averaged values of all mono-scale *RD* and *CC* maps were used to examine spatial similarity between observations and model simulations.

$$RD = (\bar{x} - \bar{y}) / \bar{x} * 100 \quad (1)$$

$$CC = \frac{1}{N^2} \sum_{i=1}^N \sum_{j=1}^N \frac{(x_{ij} - \bar{x}) \times (y_{ij} - \bar{y})}{\sigma_x \times \sigma_y} \quad (2)$$

$$\sigma_x^2 = \frac{1}{N^2 - 1} \sum_{i=1}^N \sum_{j=1}^N (x_{ij} - \bar{x})^2 \quad (3)$$

Where \bar{x} and \bar{y} represent averaged values of modelled and observational SOCD over moving windows, respectively; x_{ij} and y_{ij} are the pixel value at row i and column j of the two moving windows for the compared soil carbon stock maps. σ_x and σ_y are the standard deviations calculated within the two moving windows. The related text for the CMP method has been added in the Method section of the updated manuscript (lines 503-517, pages 22-23).

We found that eight out of eleven models, particularly the CLASS-TEM, CLAM4VIC, GTEC and SiBCASA, clearly underestimated the Tibetan soil carbon stock (**Figure R2**). Even the models such as LPJ-WSL and TEM6, which simulated a total soil carbon stock comparable to the observations, still failed to capture the spatial distribution of soil carbon stock, as indicated by the low degree of spatial agreement (low correlation values) (**Fig. R1b**; **Fig R3**). For these two models, the overestimation in the western Tibetan Plateau effectively compensates for the underestimation in the east, leading to a general model-data match in terms of the total soil carbon stock (**Fig. R2**). We have added the following text in the Results section in the updated manuscript: *“We further calculated the spatially explicit indices relative distance (RD) and cross correlation (CC) at multiple scales to examine spatial similarities between the models and our estimate using the Comparison Map Profile (CMP) method². We found that eight out of eleven models, particularly CLASS_TEM, CLAM4VIC, GTEC and SiBCASA, clearly underestimated the Tibetan soil carbon stock (Supplementary Fig. 8). We also observed a large spatial inconsistency between simulated soil carbon stock and our estimates, with a low correlation in most parts of the study area (Supplementary Fig. 9). Even the models such as LPJ-WSL and TEM6, which simulated a total soil carbon stock of comparable size to our estimate, still fail to capture the spatial distribution of soil carbon stock, as indicated by the generally low values of correlation (Supplementary Fig. 9). For these two models, the overestimation in the western Tibetan Plateau effectively compensates for the underestimation in the east, leading to an apparent model-observation match in terms of the total soil carbon stock (Supplementary Fig. 8)”* (line 220-233, page 10-11).

[Comment 26] Line 217-219: *Perhaps lack of inclusion of paleo-climate temperature could have led to some of the biases for ecosystem models, but what about discussing some of the other mechanistic assumptions with the models themselves or the soil characteristic maps that*

go into them? For example turnover time of soil carbon pools etc.? What is different between LPJ-WSL and TEM6 such that they overestimate the SOCD, where other models are much below? The distribution of models do encapsulate the observed SOCD so perhaps this can be investigated further. Also are all models simulating the same soil layer depth such that SOCD can be used universally across models— such as what you have been doing through the entire manuscript?

[Response] We strongly agree with the reviewer that other factors, besides the neglect of paleoclimate in model spin-up, such as soil carbon turnover time, could also have led to the underestimation of the Tibetan soil carbon stock by the ecosystem models. First, poor model representation of long turnover time of deep soil carbon especially in permafrost-affected regions may lead to significant underestimation of soil carbon stock^{6, 7}. Second, several typical alpine vegetation types such as marsh meadow and alpine meadow, which were characterized by a high organic carbon density and slow soil carbon turnover rate, were not well represented in the models of the MsTMIP protocol¹⁰. The details can be found in our response to **[Comment 2]**.

As stated by the reviewer, there is a substantial variation in the models' performance in simulating Tibetan soil carbon stock. For example, the models such as LPJ-WSL and TEM6 simulated a value of soil carbon stock that is statistically close to our estimate, but other models such as CLASS-CTEM and CLM4VIC showed considerable underestimation. This wide range of soil carbon stock simulation by the ecosystem models could be related to the models' different representations of soil carbon input (as indicated by NPP) and turnover time of the soil carbon pool (as given by the total soil carbon stock divided by soil heterotrophic respiration). For instance, the mean NPP (0.28 Pg C yr⁻¹) simulated by models with the lower soil carbon stock, such as CLASS-CTEM and CLM4VIC, is much lower than the mean simulated NPP (0.44 Pg C yr⁻¹) from the models with higher soil carbon stock, such as LPJ-wsl and TEM6. Moreover, the mean modeled turnover time from CLASS-CTEM and CLM4VIC (12 years, mean for the two models) is much less than that from LPJ-wsl and TEM6 (234 years, mean for the two models). We have added the following text into Supplementary Discussion of the revised manuscript to clarify these points “...*the LPJ-WSL and TEM6 models simulated a soil carbon stock that is statistically close to our estimate, but*

other models, such as CLASS-CTEM and CLM4VIC, showed a considerable underestimation (Figure 4). This wide range of soil carbon stock simulation by ecosystem models could be related to differences in the models' representation of soil carbon input (as indicated by NPP) and the turnover time of the soil carbon pool (given by the total soil carbon stock divided by soil heterotrophic respiration). For instance, the mean NPP ($0.28 \text{ Pg C yr}^{-1}$) simulated by models with the lower soil carbon stock such as CLASS-CTEM and CLM4VIC, is much lower than the mean simulated NPP ($0.44 \text{ Pg C yr}^{-1}$) from models with higher soil carbon stock such as LPJ-wsl and TEM6. Moreover, the mean modelled turnover time derived from CLASS-CTEM and CLM4VIC (12 years, mean for the two models) is much less than that derived from LPJ-wsl and TEM6 (234 years, mean for the two models)".

In addition, the soil depth tends to vary across models^{10, 16}. But for models such as VEGAS and GTEC, there is no actual information on the soil layer depth (**Table R1**). It appeared that the soil depth should not be a major source of model spread in the simulation of soil carbon stock, since there is no significant correlation between simulated soil carbon stock and soil depth across models ($r^2 = 0.07$, $P = 0.25$; **Figure R8**). For instance, the soil carbon stock derived from the CLASS-CTEM model for 4.1 m-depth (1.06 Pg C) is significantly lower than that of LPJ-WSL 1.5 m-depth outputs (37.03 Pg C) (**Table R1**).

To resolve your concern, we provided more information on the soil depth for each model in the supplementary information as Supplementary Table 2. The following text was also added into the Supplementary Discussion section: "*In addition, the soil layer depth tends to vary across models (Supplementary Table 3), but it appears that the soil depth is not a major source of the model spread in the simulation of soil carbon stock, since there was no significant correlation between the simulated soil carbon stock and soil depth across 11 models ($r^2 = 0.07$, $P = 0.25$; Supplementary Fig. 13)*".

[Comment 27] Line 230-232: *It seems odd to mention SOCD variation with depth here in the conclusions and contrast that to non-permafrost regions, when this was not mentioned in the main paper --- at least was not a focus.*

[Response] Following your kind suggestion, we have removed the comparison to the non-permafrost region in the revised manuscript.

[Comment 28] Line 236: “Our new estimate of the carbon pool, obtained by including paleoclimate as an additional predictor, is double the size of current modelled values. Future modelling of soil carbon cycling should include paleoclimate information during the model spin-up period so as to accurately represent the impacts of paleoclimate on soil properties.” So this seems like a major leap in conclusions. First off, how confident are you in the estimate of your Tibetan soil carbon stock value and SOCD? Clearly there is a limited number of sites, and a limited number of deep sites, so do you have an estimate of uncertainty based upon your site measurement approach (Figure 4)? It is likely that the uncertainty in the carbon stocks overlaps and is statistically indistinguishable from at least 2-3 models and likely as many as 5 of the ecosystem models. Looking at Figure 4, it looks like the SOCD average is dragged down by just a handful of models which simulate barely any SOCD at all including CLASS-CTEM, CLM4VIC and SIBCASA. What is going on with those models? I think a more nuanced discussion of uncertainty and also some mechanistic explanation in why there is so much variation between ecosystem models, would be more insightful.

[Response] Following your suggestion, we have made a thorough revision following major changes: **First**, we provided an uncertainty estimate of the soil carbon stock originated from sampling sites and vertical interpolation of soil carbon stock using bootstrapping combined with the Monte Carlo technique. The Tibetan soil organic carbon pool to a depth of 3 m was estimated to be 36.6 PgC (95% confidence range: 34.2 - 38.9 PgC). **Second**, we compared our updated estimate of Tibetan soil carbon stock with weighted model simulations using the method of Bayesian model averaging. In addition, a spatially explicit assessment of model-observation mismatch using the comparison map profile method is also provided. These analyses suggested that model underestimation is not due to a few extreme low simulations, but rather a general problem faced by ecosystem models in simulating soil carbon stock. **Third**, we added more mechanistic explanation associated with soil carbon turnover time, in addition to paleoclimate, such as the inadequate consideration of deep soil carbon especially in permafrost-affected regions and the poor representation of key alpine vegetation within models, which could also be responsible for the overall underestimation by models. For the details, please see the response to **[Comment 2]**.

We did observe significant variability in the performance of the ecosystem models, which could be related to the poor representation of soil carbon inputs and soil carbon turnover time in the models. We have added more explanation in this regard, for details please see the response to the above comment.

[Comment 29] *Also are paleo-climate data even available in model format? Precipitation and temperature are likely available but models require things like relative humidity, long and shortwave radiation etc. Models also function on sub-daily timesteps so that there needs to be a way to downscale coarse paleo climate data to something that is model-ready.*

[Response] As the reviewer pinpointed, besides temperature and precipitation, other paleoclimatic variables such as humidity, shortwave and longwave radiation are also necessary for modeling permafrost carbon stock in a land surface model. At present, the available paleoclimate data, including precipitation and temperature, and other variables such as wind speed, pressure and cloud cover at the monthly time step and a spatial resolution of 1 degree, are publicly accessible at <http://www.cesm.ucar.edu/experiments/cesm1.0/>.

The forcing variables required by models could be inferred from archived variables, for instance, the incoming shortwave radiation can be estimated from cloud cover. In addition, land surface models typically use the relatively high temporal resolution (e.g. daily to sub-daily) time step for process simulations. A downscaling of the relatively coarse paleoclimate variables to high temporal (e.g. sub-daily) resolution is also required. In reality, there is a growing body of studies that has developed approaches to downscale temporally coarse model outputs (e.g. monthly) to fine time scales (e.g. sub-daily)¹⁷. To resolve your concern, we have added the following text into the revised MS: “*Additionally, to enable future land surface models to fully account for paleoclimatic impacts on permafrost soil carbon stock, other paleoclimatic variables such as humidity, pressure and radiation, not just temperature and precipitation, should be prepared and downscaled to the relatively high temporal resolution (e.g. sub-daily) required by the models*” (line 311-315, page 14).

Response to Reviewer #2

[Comment 1] The authors apply statistical analysis methods to analyze the relationship between permafrost distribution (and various soil properties) data collected in the field, and climate indicators derived from data (present-day) and model outputs (mid Holocene and LGM). They also provide a new estimate for total soil carbon in the Tibetan permafrost region and compare this to recent modelling efforts.

A main finding is that modelling the permafrost region of Tibet using only present-day climate conditions results in an underestimate of total carbon in those soils (and that this conclusion may be extended to other permafrost regions).

Overall this is an interesting and important study, quantifying the effect of neglecting paleoclimate forcing in the modelling of permafrost carbon stock. It makes extensive use of correlating large datasets, and it looks like a lot of work has gone into the study. It relies heavily on correlation statistics, but this is probably an appropriate method, and those results should certainly be reproducible with the datasets.

It would nice to have some more discussion on the physical mechanisms that result in paleo climate imprint on existing permafrost soils, with a schematic in the style of figure s3, to clarify the argument.

[Response] Many thanks for your insightful and positive comments that have helped us to improve our manuscript. Following your constructive suggestions, we have added a schematic diagram (**Figure R9**) and more discussion to clarify the mechanisms that lead to a detectable paleoclimatic footprint in modern permafrost soil carbon stock. Details can be seen in the responses to the following comments.

Figure R9 Schematic of paleoclimate legacy on present-day soil carbon stock through direct paleo-carbon deposit or indirectly changing soil physiochemical properties.

[Comment 2] Line 177: It would be useful to clarify what is the indirect mechanism by which soil properties affect organic carbon stabilisation (is this chemical, mechanical?)

[Response] The indirect impact of paleoclimate on current soil carbon stock operated through changing soil physical (e.g. soil texture) and geochemical properties (e.g. cation exchange capacity, total phosphorus and potassium). The soil physiochemical characteristics, which have slowly evolved under the influence of past climate regimes, determine the soil capacity to stabilize soil carbon inputs and then constitute a mechanism for a possible paleoclimatic footprint on present-day soil carbon stock^{18, 19, 20, 21}. Moreover, there is increasing evidence to show the importance of physiochemical properties in controlling soil carbon stock over the Tibetan Plateau^{22, 23}. To address the reviewer’s comment, we have incorporated the following text into the revised manuscript: *“The soil physiochemical characteristics (i.e. soil texture, cation exchange capacity, total phosphorus and potassium), determining the capacity to stabilize soil carbon inputs^{18, 19, 20}, have evolved slowly under the influence of past climate regimes, and there is increasing evidence showing the importance of physiochemical*

properties in controlling soil carbon stock over the Tibetan Plateau^{13, 22},” (lines 135-140, page 7).

[Comment 3] Line 133: You say the ¹⁴C indicates that the *distribution* of permafrost is largely a relic of that formed 35ka to 10.8ka BP. Does line 133 to 135 then suggest very little re-advance of permafrost since the mid Holocene?

[Response] Based on the evidence from relic permafrost and periglacial phenomena, most of the Tibetan permafrost was formed during the Last Glacial Maximum (35ka to 10.8ka BP), when the lower elevation limit of permafrost was 1000 m lower than today²⁴. Intensive permafrost degradation subsequently occurred during the Holocene Thermal Maximum, with the lower elevation limit of permafrost ~300-400 m higher than today and the total permafrost area ~50-60% less than today. So, the lower elevation limit of permafrost has shifted upward by 1300-1400 m from LGM to Mid-Holocene. After the mid-Holocene, re-advance of the permafrost did occur during the little ice age (LIA), but only to a lesser degree, with the lower elevation limit being only 150-200 m lower than today^{25, 26}. The re-advance of permafrost during LIA stays well within the relics of permafrost that formed during the LGM. The main body of the Tibetan permafrost was thus formed during the LGM, which is the most important climatic epoch for defining the present-day spatial pattern of relic permafrost.

In addition, the compilation of radiocarbon age data from relic permafrost (¹⁴C) over the Tibetan Plateau showed that the carbon age can be dated back to 20403-39830 years BP^{24, 27}, supporting the direct paleoclimatic effect on soil carbon stock. We have rewritten this sentence as follows “*This observation is in agreement with the results of soil radiocarbon (¹⁴C) dating studies in the relic permafrost, where the carbon age can be dated to 20~40 thousand years BP^{24, 27}*” (line 130-132, page 7).

[Comment 4] Line 140: This paragraph states that during the mid-Holocene there was probably a deepening of the active layer, and this assumption is used to support a conclusion that soil properties are probably the controller of paleoclimate legacies on soil carbon stocks. I think this point needs to be expanded. Is it reasonable to assume that the active layer deepened a lot at the mid-Holocene? How would this manifest in your data? Frost index has

*been used as an indicator for permafrost extent, and that is sensitive to seasonal insolation. 6ka was in a falling Northern hemisphere summer insolation period, affecting the freezing and thawing cycle, how does the temperature seasonality variable change since the NH summer insolation maximum (~11ka) to the mid-Holocene?. Perhaps there was not a *large* top-down deepening of the active layer, and how would this affect your interpretation?*

[Response] To understand whether there was intensive permafrost thawing at the mid-Holocene, we calculated the freezing index (*FI*, Equation 4-5) for LGM and MidH periods using paleoclimate simulations from the CESM model, and analyzed the relative change of the freezing index from LGM to MidH to infer the change of permafrost extent. The freezing index was calculated by accumulating the average daily temperatures below 0 °C²⁸.

$$FI = \sum_{i=1}^N |T_i|, T_i < 0^{\circ}\text{C} \quad (4)$$

$$FI \text{ change (\%)} = (FI_{\text{LGM}} - FI_{\text{MidH}})/FI_{\text{LGM}} * 100 \quad (5)$$

We found a decrease of the freezing index from LGM to MidH, with a mean magnitude of 22% (**Figure R10**). This decrease in the freezing index is 4 times that from 1958 to 2002²⁹, during which period the mean annual air temperature increased by approximately 1.5 °C. It thus confirmed that intensive permafrost degradation did occur during the Mid-Holocene period. Permafrost relicts, such as relict permafrost tables, thawed sandwiches (taliks) and periglacial phenomena could provide further evidence for this result. Previous studies have shown that the Tibetan permafrost thawed downwards to a depth of 15-25 m, with the lower elevation limit of permafrost shifted upwards by 1300-1400 m since the LGM^{24, 25, 26}.

The intensive mid-Holocene thawing would substantially increase the decomposition of soil carbon that accumulated during the LGM period especially for shallow soil layers, and indirectly support the hypothesis that soil properties are the main controller of paleoclimate legacies on the Tibetan soil carbon stock. This argument is also supported by our structure equation analysis (SEM) and partial correlations. We showed that the indirect effect (standardized effect = 0.39) is larger than the direct effect (standardized effect = 0.30). Moreover, the correlations between soil carbon and modern climate variables significantly

decreased, or even became insignificant, after removing the effects of soil properties.

In the revised manuscript, we have added more discussion on the mid-Holocene thawing as follows: “*During the Holocene Thermal Maximum (HTM), intensive permafrost thawing occurred down to a depth of ~15-25 m on the Tibetan Plateau^{24, 26}, and the areal extent of permafrost represented by the freezing index (see Methods) also decreased substantially, with the magnitude of the mean decline being nearly 22% over the Tibetan Plateau (Supplementary Fig. 4). Such thawing would be expected to have greatly increased soil carbon decomposition*” (lines 149-155, pages 7-8).

Figure R10 Spatial pattern of the relative change of freezing index (FI) from Last Glacial Maximum to mid-Holocene.

[Comment 5] Line 144: Is the top 10cm of soil actually representative of the top 30cm (which is the basis of correlations in figure 2)? If the same accumulation rates hold until 30cm (which is probably not the case), these soils would be older than mid-Holocene age. What do you consider to be “paleo-vegetation”? I assume you mean LGM-age vegetation. Why do you not have ¹⁴C dating up to 30cm (at least in some places)? Is it possible that actually the soils at 30cm are already LGM age?

[Response] To gain further knowledge on the radiocarbon age for the soil layer up to 30 cm, we expanded the literature survey of ¹⁴C data over the Tibetan Plateau (**Table R2**), and found that soil carbon could be dated back to ~3.3 ka for 10 cm depth and up to ~8.9 ka for 30 cm

depth. It is therefore possible, but with very low probability, that soil carbon of 30 cm depth originated from mid-Holocene. Nonetheless, there is no evidence to support the view that the carbon at 30 cm depth originated from the LGM age.

Therefore, to err on the side of caution, the paleo-vegetation here should mean the LGM-age vegetation, as the reviewer suggested. We have revised the sentence as follows: “*The indirect mechanism is further supported by radiocarbon dating of soil organic carbon in the soils over the Tibetan Plateau (Supplementary Table 2). The relatively young age obtained by this method suggests that the current top soil layers are not likely to originate from LGM-vegetation*” (line 158-161, page 8).

Table R2 The soil carbon age data from soil radiocarbon (^{14}C) dating studies on the Tibetan Plateau

Soil depth (m)	Age (yr BP)	Location	Reference
0.105	> 1958 AD	31.32°N, 92.06°E	Kaiser et al. , 2008 ³⁰
0.15	>1958 AD	31.59°N, 91.48°E	Kaiser et al. , 2008 ³⁰
0.1	>1958 AD	31.32°N, 92.06°E	Kaiser et al. , 2008 ³⁰
0.14	723	31.86°N, 93.09°E	Kaiser et al. , 2008 ³⁰
0.17	1882	31.76°N, 92.63°E	Kaiser et al. , 2008 ³⁰
0.14	395	31.32°N, 92.06°E	Kaiser et al. , 2008 ³⁰
0.29	798	31.59°N, 91.48°E	Kaiser et al. , 2008 ³⁰
0.18	321	Not available	Kaiser et al. , 2008 ³⁰
0.25	8894	37.54°N, 101.33°E	Kaiser et al. , 2007 ³¹
0.1	3295	32.98°N, 98.10 °E	Jin et al. , 2007 ²⁴
0.2	3925	34.48°N, 100.23 °E	Jin et al. , 2007 ²⁴
0.3	3270	30.07°N, 90.55°E	Jin et al. , 2007 ²⁴
0.4	7207	35.88°N, 94.50°E	Jin et al. , 2007 ²⁴
0.5	1080	34.91°N, 97.71°E	Jin et al. , 2007 ²⁴
0.3	3270	30.08°N, 90.55°E	Jin et al. , 2007 ²⁴

[Comment 6] Line 190: There does not appear to be an uncertainty value on the total estimated carbon stock for Tibet permafrost derived in this study, which would be useful.

[Response] Following your suggestion, we have assessed the uncertainties associated with the sampling sites and the vertical profile extrapolation by using bootstrapping and Monto Carlo

methods. **To account for the uncertainty introduced by sampling sites,** we adopted a bootstrap method (random sampling with replacement) to generate 1,000 pseudo replicates, which were used to establish the SVM model in estimating the 30 cm soil carbon stock. Here we relied on a regression model for vertical extrapolation of deeper-layer soil carbon stock from the top 30 cm layer. The regression model was derived from the relationship between soil organic carbon density and soil depth across 325 sites with deep soil profile data. **To estimate the uncertainty due to the vertical extrapolation,** we adopted the Monte Carlo sampling technique to draw 1,000 random sets of predicted values from their normal distributions with the estimated mean and standard deviations for each grid. These two types of uncertainty were then merged to yield an estimate of the uncertainty in the Tibetan soil carbon stock. Our analysis showed that the Tibetan soil organic carbon pool to a depth of 3 m was 36.6 PgC (95% confidence range: 34.2 - 38.9 PgC), and the mean soil organic carbon density was estimated to be 15.4 kg C m⁻² (95% confidence range: 14.4-16.4 kg C m⁻²). To resolve your concern, we have added the following text into the Methods section: “*An uncertainty estimate that originated from sampling sites and vertical interpolation of soil carbon stock was provided for the estimation of soil carbon stock on the Tibetan Plateau. To account for the uncertainty introduced by sampling sites, we adopted a bootstrap method (random sampling with replacement) to generate 1,000 pseudo replicates, which were used to establish the SVM model in estimating the 30 cm soil carbon stock. Here we relied on a regression model for vertical extrapolation of deeper-layer soil carbon stock from the top 30 cm layer. The regression model was derived from the relationship between soil organic carbon density and soil depth across 325 sites with deep soil profile data. To estimate the uncertainty due to the vertical extrapolation, we adopted the Monte Carlo sampling technique to draw 1,000 random sets of predicted values from their normal distributions with the estimated mean and standard deviations obtained from the regression model for each grid. These two types of uncertainty were merged to yield an estimate of the uncertainty in Tibetan soil carbon stock*” (lines 479-492, page 21-22). In addition, the numbers related to the Tibetan soil carbon stock and soil carbon density were also updated accordingly throughout the manuscript.

Response to Reviewer #3

[Comment 1] The paper describes an innovative and novel approach to study climate-driven past permafrost soil processes and consequent inherited legacies, which led to formation of current carbon stock in Tibetan mountains. The authors conclude that representative palaeoclimate data combined with information of soil physicochemical properties yields more reliable carbon stock estimations, than if the palaeoclimate reconstruction data is not properly applied, i.e. when ecosystem modelling only accounts for the pre-industrial climate conditions. The new carbon stock estimate doubled the size of the C storage.

Basically, I have only positive things to say. The paper is very well written and the study description proceeds logically. The interest is not lost, despite various steps are taken and a huge set of tools and study components is described and used. It is clear that the authors have used lot of time and energy in trying identifying the most suitable ways to understand relationship between (palaeo)climate and past and current (and future) carbon dynamics. The study is also exploiting extensive field data collected over the years. I am very happy to see palaeoclimate aspect highlighted in understanding current and future C processes.

Undoubtedly, extrapolations and generalisations used in this study infold uncertainties and some assumptions, for instance soil depths and stored C, have to be considered tentative. However, I felt that the authors were aware of the limitations. My background is not in modeling but I found it very interesting to read how various models were step-by-step applied to quantify different variables and in-between linkages. A respectable amount of different data-bases were exploited. The study is local/regional and as such does not have global implications. (I leave it to the editors to decide if the novelty of study method is enough to allow publication in Nature Communications, or would a more traditional palaeo journal, such as QSR, be a better forum). In any case, as the carbon stock estimate quite radically changed for Tibetan permafrost area when palaeoclimate was better accounted, the approach should be further applied to other regions from where C stock model estimations exist. If similar increase in stock values is repeated, larger-scale model revisions are needed.

[Response] We are very grateful to the reviewer for the professional comments. With respect to potential concerns raised by the reviewer, we have done a thorough revision on the

manuscript.

First, in terms of uncertainties related to extrapolations of Tibetan soil carbon stock we have provided an uncertainty estimate of Tibetan soil carbon stock associated with sampling sites and vertical interpolation. After considering these two sources of uncertainties, the Tibetan soil organic carbon pool to a depth of 3 m was 36.6 PgC (95% confidence range: 34.2 - 38.9 PgC), and the mean soil organic carbon density was estimated to be 15.4 kg C m⁻² (95% confidence range: 14.4-16.4 kg C m⁻²). The following text was also added into the Method section of the revised manuscript: “*An uncertainty estimate that originated from sampling sites and vertical interpolation of soil carbon stock was provided for the estimation of soil carbon stock on the Tibetan Plateau. To account for the uncertainty introduced by sampling sites, we adopted a bootstrap method (random sampling with replacement) to generate 1,000 pseudo replicates, which were used to establish the SVM model in estimating the 30 cm soil carbon stock. Here we relied on a regression model for vertical extrapolation of deeper-layer soil carbon stock from the top 30 cm layer. The regression model was derived from the relationship between soil organic carbon density and soil depth across 325 sites with deep soil profile data. To estimate the uncertainty due to the vertical extrapolation, we adopted the Monte Carlo sampling technique to draw 1,000 random sets of predicted values from their normal distributions with the estimated mean and standard deviations obtained from the regression model for each grid. These two types of uncertainty were merged to yield an estimate of the uncertainty in Tibetan soil carbon stock.*” (lines 479-492, pages 21-22).

Second, although this study was conducted at the regional scale, the Tibetan Plateau holds the largest permafrost area outside the polar regions and is home to the largest area of alpine permafrost in the world^{13,32}. Revealing the possible paleoclimate-permafrost soil carbon relationship and its underlying mechanisms has substantial implications for understanding alpine permafrost soil carbon dynamics in a future warmer world. Our general findings about paleoclimate legacies on permafrost soil carbon stock, and their increasing importance with soil depth, could be safely generalized to arctic and boreal permafrost

regions, because of common frozen environments, high soil carbon concentration and slow soil carbon turnover rate. Furthermore, we have provided evidence to indicate that the impact of paleoclimate on present-day soil carbon stock (particularly for shallow soil layers) was mainly through changing soil physical (e.g. soil texture) and geochemical properties (e.g. cation exchange capacity, total phosphorus and potassium). It helps reconcile the apparent conflict between the relatively young age of organic carbon in shallow soil layers indicated by radiocarbon measurements and the statistically-inferred paleoclimatic fingerprint in previous studies³³.

But as the reviewer correctly pointed out, it is still highly necessary to apply the approach of detecting the paleoclimatic footprint on soil carbon stock to other permafrost regions, such as the pan-arctic region. On the one hand, there are regional differences with respect to the extent and intensity of paleoclimatic change, and these could mean that the relative importance of paleoclimates during LGM and mid-Holocene to permafrost soil carbon stock could vary between the pan-arctic region and the Tibetan Plateau. Moreover, to what extent the paleoclimate legacies on permafrost soil carbon can be attributed to the influence of paleoclimate on soil physiochemical properties remains uncertain. On the other hand, how much ecosystem models would underestimate permafrost soil carbon stock, and to what degree this underestimation could be attributed to the neglect of paleoclimate in the model spin-up, needs to be resolved over the pan-arctic region. To address potential concerns raised by the reviewer, we have added the following discussion into the text: “*In addition, the methodology introduced in this paper could be used to quantitatively assess the paleoclimatic fingerprint on the permafrost soil carbon stock in other permafrost regions such as the pan-arctic region. Such assessments could provide a more complete understanding of paleoclimate effects on permafrost soil carbon stock that in turn can help with the understanding of permafrost soil carbon dynamics in a warmer future*” (lines 296-302, pages 13-14).

In addition, in the revised manuscript, we have replaced the original arithmetic model average approach with the Bayesian model averaging method, which is conditional on independent observation data¹, to diagnose model deficiencies associated

with the neglect of paleoclimate. We highlighted the use of Bayesian model averaging method in quantifying model-data mismatch over the pan-arctic region, since this could tone down the contribution of models that have notable deficiencies in the accurate simulation of major physiological processes such as photosynthesis, and help us to identify model errors associated with the neglect of the effect of paleoclimate. Accordingly, we have added the following text in the revised manuscript: “*Here we used the Bayesian Model Averaging method (BMA), which is conditional on an independent observation data¹, to tone down the role of models that have notable deficiencies in representing major physiological processes. Given the availability of satellite-derived net primary productivity (NPP) product⁵, we adopted NPP, as an indicator of vegetation carbon input, to rank the model performance. Larger weights were assigned to models that have a better performance in simulating NPP with respect to satellite-derived observations*” (lines 208-214, page 10).

[Comment 2] One annoying detail: “megathermal”, established term is Holocene Thermal Maximum HTM

[Response] Thanks for your suggestion. We have changed “megathermal” into “Holocene Thermal Maximum” in the revised manuscript.

References

1. Wasserman L. Bayesian Model Selection and Model Averaging. *J. Math. psychol.* **44**, 92-107 (2000).
2. Gaucherel C, Alleaume S, Hely C. The Comparison Map Profile Method: A Strategy for Multiscale Comparison of Quantitative and Qualitative Images. *IEEE T. Geosci. Remote* **46**, 2708-2719 (2008).
3. Huntzinger DN, *et al.* The North American Carbon Program Multi-Scale Synthesis and Terrestrial Model Intercomparison Project – Part 1: Overview and experimental design. *Geosci. Model Dev.* **6**, 2121-2133 (2013).
4. Todd-Brown KEO, *et al.* Causes of variation in soil carbon simulations from CMIP5 Earth system models and comparison with observations. *Biogeosciences* **10**, 1717-1736 (2013).
5. Kolby Smith W, *et al.* Large divergence of satellite and Earth system model estimates of global terrestrial CO₂ fertilization. *Nat. Clim. Change* **6**, 306 (2015).
6. Koven CD, *et al.* The effect of vertically resolved soil biogeochemistry and alternate soil C and N models on C dynamics of CLM4. *Biogeosciences* **10**, 7109-7131 (2013).
7. Luo Y, *et al.* Toward more realistic projections of soil carbon dynamics by Earth system models. *Glob. Biogeochem. Cy.* **30**, 40-56 (2016).

8. Schuur EAG, *et al.* Climate change and the permafrost carbon feedback. *Nature* **520**, 171-179 (2015).
9. Strauss J, *et al.* Deep Yedoma permafrost: A synthesis of depositional characteristics and carbon vulnerability. *Earth-Sci Rev.* **172**, 75-86 (2017).
10. Tian H, *et al.* Global patterns and controls of soil organic carbon dynamics as simulated by multiple terrestrial biosphere models: Current status and future directions. *Glob. Biogeochem. Cy.* **29**, 2014GB005021 (2015).
11. Dutta K, Schuur EAG, Neff JC, Zimov SA. Potential carbon release from permafrost soils of Northeastern Siberia. *Glob. Change Biol.* **12**, 2336-2351 (2006).
12. Zimov SA, Schuur EAG, Chapin FS. Permafrost and the Global Carbon Budget. *Science* **312**, 1612-1613 (2006).
13. Mu C, *et al.* Editorial: Organic carbon pools in permafrost regions on the Qinghai-Xizang (Tibetan) Plateau. *Cryosphere* **9**, 479-486 (2015).
14. Grace JB. *Structural Equation Modeling and Natural Systems*. Cambridge University Press (2006).
15. Shipley B. *Cause and Correlation in Biology*. Cambridge University Press (2000).
16. McGuire AD, *et al.* Variability in the sensitivity among model simulations of permafrost and carbon dynamics in the permafrost region between 1960 and 2009. *Glob. Biogeochem. Cy.*, 2016GB005405 (2016).
17. Lee T, Park T. Nonparametric temporal downscaling with event-based population generating algorithm for RCM daily precipitation to hourly: Model development and performance evaluation. *J. Hydrol.* **547**, 498-516 (2017).
18. Fontaine S, Barot S, Barre P, Bdioui N, Mary B, Rumpel C. Stability of organic carbon in deep soil layers controlled by fresh carbon supply. *Nature* **450**, 277-280 (2007).
19. Six J, Conant RT, Paul EA, Paustian K. Stabilization mechanisms of soil organic matter: Implications for C-saturation of soils. *Plant Soil* **241**, 155-176 (2002).
20. Doetterl S, *et al.* Soil carbon storage controlled by interactions between geochemistry and climate. *Nat. Geosci.* **8**, 780-783 (2015).
21. Doetterl S, *et al.* Links among warming, carbon and microbial dynamics mediated by soil mineral weathering. *Nat. Geosci.* **11**, 589-593 (2018).
22. Fang K, Qin S, Chen L, Zhang Q, Yang Y. Al/Fe Mineral Controls on Soil Organic Carbon Stock Across Tibetan Alpine Grasslands. *J. Geophys. Res-Bioge.* **124**, 247-259 (2019).
23. Mu CC, *et al.* Soil organic carbon stabilization by iron in permafrost regions of the Qinghai-Tibet Plateau. *Geophys. Res. Lett.* **43**, 2016GL070071 (2016).
24. Jin HJ, Chang XL, Wang SL. Evolution of permafrost on the Qinghai-Xizang (Tibet) Plateau since the end of the late Pleistocene. *J. Geophys. Res.* **112**, F02S09 (2007).
25. Wang S, Huijun J, Shuxun L, Lin Z. Permafrost degradation on the Qinghai-Tibet Plateau and its environmental impacts. *Permafrost and Periglacial Processes* **11**, 43-53 (2000).
26. Jin H, Zhao L, Wang S, Jin R. Thermal regimes and degradation modes of permafrost along the Qinghai-Tibet Highway. *Science in China Series D: Earth Sciences* **49**, 1170-1183 (2006).
27. Cheng J, X. , J. Zhang, M. Z. Tian, W. Y. Yu, D. X. Tang, Yue JW. Ice-wedge casts

- discovered in the source area of the Yellow River, northeast Tibetan Plateau and their paleoclimatic implications. *Quat. Sci.* **26**, 92-98 (2006).
28. Frauenfeld OW, Zhang T, Mccreight JL. Northern Hemisphere freezing/thawing index variations over the twentieth century. *Int. J. Climatol.* **27**, 47-63 (2007).
 29. Frauenfeld OW, Zhang T, Mccreight JL. Northern Hemisphere freezing/thawing index variations over the twentieth century. *Int. J. Climatol.* **27**, 47-63 (2007).
 30. Kaiser K, *et al.* Turf-bearing topsoils on the central Tibetan Plateau, China: Pedology, botany, geochronology. *Catena* **73**, 300-311 (2008).
 31. Kaiser K, Schoch WH, Mieke G. Holocene paleosols and colluvial sediments in Northeast Tibet (Qinghai Province, China): Properties, dating and paleoenvironmental implications. *Catena* **69**, 91-102 (2007).
 32. Zhang T, Barry RG, Knowles K, Heginbottom JA, Brown J. Statistics and characteristics of permafrost and ground-ice distribution in the Northern Hemisphere. *Polar Geography* **31**, 47-68 (2008).
 33. Delgado-Baquerizo M, *et al.* Climate legacies drive global soil carbon stocks in terrestrial ecosystems. *Sci. Adv.* **3**, e1602008 (2017).

REVIEWERS' COMMENTS:

Reviewer #1 (Remarks to the Author):

This reviewer appreciated the detailed and professional response to my original concerns and suggestions with the first version of the manuscript. The authors have addressed my concerns related to a more thorough discussion of how the 11 terrestrial ecosystem models compare to the regional carbon stock product. They now provide an uncertainty estimate for the regional carbon stock product, and provide a model Bayesian averaged carbon stock estimate to demonstrate that the model underestimation of carbon is systemic. They also provide more discussion about the reasons for this underestimation including deficiencies in the model spinup, as well as potential problems with carbon turnover times. More thorough spatial maps are provided to compare model vs. observed carbon stock performance. This reviewer recommends this manuscript for publication. Please see some remaining minor comments below:

Line 1:carbon stock of 'the' Tibetan permafrost region

Line 51: Problem with uncertainty formatting. Also should be 'triple' not 'treble'. Also what is 's.e.m.' ??

Line 112: replace 'since' with 'because'

Line 115: Replace 'under' with 'during'

Line 131: Not sure why in some cases the authors use 'relic' and other times 'relict' is used.

Line 214-233: A very nice way to address model performance as it compares to the observations.

Line 234- 237: Need to state here what was done to 'connect' the model spinup with the present day carbon stocks. It is implied that a transient simulation was run between early 1900's to present day, but that needs to be stated explicitly here – can't remember if this is stated earlier in manuscript.

Also should probably state some criteria that you used to evaluate whether spinup was complete. Did it come to equilibrium?

Line 294: Again triple, not treble.

Line 294 – 296: “Future modelling of soil carbon cycling should include paleoclimate information during the model spin-up period so as to accurately represent the impacts of paleoclimate on soil properties.”

That’s a nice idea but the models I am familiar with generally prescribe soil properties, and do not allow for interaction with climate. Therefore even if paleoclimate was used – it could certainly influence the soil carbon pools, but certainly would not influence the soil properties dynamically.

Methods: Climate and Soil Property data Would be helpful here to explain to what extent this products were used to ‘drive’ the terrestrial models, or to be used as validation. Believe only modern climate data was used to perform present day simulations, not as clear cut, what, if any soil data was used as a boundary conditions for model or if it was all taken from MsTMIP methodology. AT other parts in the text you sate that pre-industrial period early 1900’s were used to spinup the model – you should state that here too.

Lines 493-502: May want to emphasize here that the period 1975-2010 is modern weather, and not really historical climate and certainly not paleoclimate. You are clear in the discussion and other parts of the manuscript to distinguish between these two things – but you may want to make that clearer in the methods too.

Figure 1: Really nice improvement to this figure.

Figure 3: The word ‘modern’ looks like ‘modem’ . Could you provide more spacing?

Figure 4: Nice improvement especially with panel a.

Supplimentary Figure 10: Probably should say ‘modern’ instead of ‘current’ to be consistent with text.

The term ‘simulation intials’ is strange and jargony.

Supplementary Figure 11: Confusing. I think this means bias in soil carbon stock based upon vegetation specific region – but this is never explicitly stated.

Reviewer #2 (Remarks to the Author):

The authors have made significant changes to their manuscript in line with the first set of reviewers comments. In my opinion it is now a more robust piece of work.

The specific items I raised have been satisfactorily addressed.

The main point of the paper I think is well argued: that paleoclimate forcing is necessary to correctly represent Tibetan permafrost carbon stocks. This has implications for other permafrost regions. The community modelling and studying present and future permafrost and carbon dynamics in these soils (in any world region) will benefit from the publication of this study, as will anyone interested in global soil carbon stocks, the global carbon budget and how models are spun-up to represent present-day conditions.

Response to Reviewer #1

[Comment 1] This reviewer appreciated the detailed and professional response to my original concerns and suggestions with the first version of the manuscript. The authors have addressed my concerns related to a more thorough discussion of how the 11 terrestrial ecosystem models compare to the regional carbon stock product. They now provide an uncertainty estimate for the regional carbon stock product, and provide a model Bayesian averaged carbon stock estimate to demonstrate that the model underestimation of carbon is systemic. They also provide more discussion about the reasons for this underestimation including deficiencies in the model spinup, as well as potential problems with carbon turnover times. More thorough spatial maps are provided to compare model vs. observed carbon stock performance. This reviewer recommends this manuscript for publication. Please see some remaining minor comments below:

[Response] Thanks again for the reviewer's insightful and positive comments. We further revised our manuscript according to the detailed comments from the reviewer. Details can be seen in the responses to the following comments.

[Comment 2] Line 1:carbon stock of 'the' Tibetan permafrost region

[Response] Done as suggested.

[Comment 3] Line 51: Problem with uncertainty formatting. Also should be 'triple' not 'treble'. Also what is 's.e.m.' ??

[Response] We have replaced "treble" by "triple". The term "s.e.m." here means "standard error", according to the protocol of the *Nature Communications*.

[Comment 4] Line 112: replace 'since' with 'because'

[Response] Done as suggested.

[Comment 5] Line 115: Replace 'under' with 'during'

[Response] Done as suggested.

[Comment 6] Line 131: Not sure why in some cases the authors use 'relic' and other times 'relict' is used.

[Response] To avoid confusion, we use “relict” consistently in the revised manuscript.

[Comment 7] Line 214-233: A very nice way to address model performance as it compares to the observations.

[Response] Thanks for your praise.

[Comment 8] Line 234- 237: Need to state here what was done to 'connect' the model spinup with the present day carbon stocks. It is implied that a transient simulation was run between early 1900's to present day, but that needs to be stated explicitly here – can't remember if this is stated earlier in manuscript.

[Response] Yes, there is a transient simulation that was run between early 1900's to present-day. Following your kind suggestion, we have added the following text into the manuscript “*The present-day soil carbon stock was derived from the transient simulation that started from steady-state initial conditions after spin-up and was run forward in time through the historical period until 2010, using observed time-varying climate and CO235. In these models, the soil carbon pools were initialized by the early 1900's climate rather than the paleoclimate*” (lines 241-245 of page 11).

[Comment 9] Also should probably state some criteria that you used to evaluate whether spinup was complete. Did it come to equilibrium?

[Response] Yes, the simulated carbon pools came to the equilibrium after the model spin-up in MsTMIP experimental design (https://nacp.ornl.gov/MsTMIP_variables.shtml)¹. The steady-state criterion for carbon fluxes is that the 100-year mean change in total ecosystem

carbon stock must be below $1 \text{ g m}^{-2} \text{ yr}^{-1}$ during the model spin-up¹. This information has been added in the Methods as follows: “*The simulated carbon pools came to the equilibrium after the model spin-up in MsTMIP experimental design (https://nacp.ornl.gov/MsTMIP_variables.shtml). The steady-state criterion for carbon fluxes is that the 100-year mean change in total ecosystem carbon stock must be below $1 \text{ g m}^{-2} \text{ yr}^{-1}$ during the model spin-up¹” (lines 528-532 of pages 22).*

[Comment 10] Line 294: Again triple, not treble.

[Response] Done as suggested.

[Comment 11] Line 294 – 296: “*Future modelling of soil carbon cycling should include paleoclimate information during the model spin-up period so as to accurately represent the impacts of paleoclimate on soil properties.*” *That’s a nice idea but the models I am familiar with generally prescribe soil properties, and do not allow for interaction with climate.*

Therefore even if paleoclimate was used – it could certainly influence the soil carbon pools, but certainly would not influence the soil properties dynamically.

[Response] We agree with the reviewer that soil properties are not allowed to interact with climate in current generation of models and thus make a dynamic estimation of soil properties a big challenge. We have therefore added above point in the Discussion as follows: “*Future modelling of soil carbon cycling should include paleoclimate as well as its interaction with soil properties during the model spin-up period so as to accurately represent the impacts of paleoclimate on soil properties*” (lines 306-309 of page 13).

[Comment 12] *Methods: Climate and Soil Property data · Would be helpful here to explain to what extent this products were used to ‘drive’ the terrestrial models, or to be used as validation. Believe only modern climate data was used to perform present day simulations, not as clear cut, what, if any soil data was used as a boundary conditions for model or if it was all taken from MsTMIP methodology. AT other parts in the text you sate that*

pre-industrial period early 1900's were used to spinup the model – you should state that here too.

[Response] Following your kind suggestion, we have added the following text in the Methods:

“The climate (both modern and paleo climate) and soil properties data were used to assess the relative importance of modern and paleoclimate in affecting modern soil carbon stock using a variety of statistical methods, and then subjected to the SVM model to predict the modern carbon stock over the entire study region. Note that the modern climate and soil property data are based on observation data sets, while paleoclimate data were retrieved from the Community Climate System Model as described above” (lines 417-423 of page 18).

In addition, more details about the model spin-up along with the references were added in the Discussion Section as follows: *“The simulated carbon pools came to the equilibrium after the model spin-up in MsTMIP experimental design (https://nacp.ornl.gov/MsTMIP_variables.shtml). The steady-state criterion for carbon fluxes is that the 100-year mean change in total ecosystem carbon stock must be below $1 \text{ g m}^{-2} \text{ yr}^{-1}$ during the model spin-up¹”* (lines 528-532 of pages 22).

[Comment 13] *Lines 493-502: May want to emphasize here that the period 1975-2010 is modern weather, and not really historical climate and certainly not paleoclimate. You are clear in the discussion and other parts of the manuscript to distinguish between these two things – but you may want to make that clearer in the methods too.*

[Response] Thanks for your constructive suggestion. We have added such information in the method section as follows: *“We compiled a climate data set composed of climate over the period 1975-2015 (modern climate), and paleo climates in the mid-Holocene (MidH) and the Last Glacial Maximum (LGM)”* (lines 377-379 of page 16).

[Comment 14] *Figure 1: Really nice improvement to this figure.*

[Response] Thanks for the praise.

[Comment 15] *Figure 3: The word ‘modern’ looks like ‘modem’ . Could you provide more*

spacing?

[Response] Done as suggested.

[Comment 16] Figure 4: Nice improvement especially with panel a.

[Response] Thanks for your praise.

[Comment 17] Supplementary Figure 10: Probably should say 'modern' instead of 'current' to be consistent with text. The term 'simulation initials' is strange and jargony.

[Response] We have replaced 'current' by 'modern' in the figure. We also revised the figure caption as follows: "**Supplementary Figure 10. Dependence of modern soil organic carbon stock simulations on initial soil carbon stocks (mean values during the 1901-1910) across the 11 ecosystem models**".

[Comment 18] Supplementary Figure 11: Confusing. I think this means bias in soil carbon stock based upon vegetation specific region – but this is never explicitly stated.

[Response] Yes. The bias here means the bias in soil carbon stock over vegetation specific regions. We added such information in the revised figure caption as follows: "**Supplementary Figure 11. Bias in soil carbon stock based upon vegetation specific region as indicated by the relative distance of weighted ensemble mean of the 11 model outputs as compared to the new observational estimate in this study. The black dots and lines inside the boxes represent mean and median values, and box ends represent 25th to 75th quartile range, and whiskers denote 10th to 90th quartile range**".

Response to Reviewer #2

The authors have made significant changes to their manuscript in line with the first set of reviewers comments. In my opinion it is now a more robust piece of work. The specific items I raised have been satisfactorily addressed. The main point of the paper I think is well argued: that paleoclimate forcing is necessary to correctly represent Tibetan permafrost carbon stocks. This has implications for other permafrost regions. The community modelling and studying present and future permafrost and carbon dynamics in these soils (in any world region) will benefit from the publication of this study, as will anyone interested in global soil carbon stocks, the global carbon budget and how models are spun-up to represent present-day conditions.

[Response] Again, thanks for the reviewer's insightful and positive comments.

Reference

- 1 Huntzinger, D. N. *et al.* The North American Carbon Program Multi-Scale Synthesis and Terrestrial Model Intercomparison Project – Part 1: Overview and experimental design. *Geosci. Model Dev.* **6**, 2121-2133, doi:10.5194/gmd-6-2121-2013 (2013).